# THE FEW GOVERN THE MANY: UNVEILING FEW-LAYER DOMINANCE FOR TIME SERIES MODELS

## ABSTRACT

Time series (TS) forecasting plays a vital role in practice, but remains a highly challenging task. The outstanding performance of Large-scale models across multiple domains has driven the advancement of Large-scale TS models, providing an effective pathway for forecasting task. **Performance degradation** has been observed in Large-scale TS models, demonstrating that **bigger is not always better**, which is a puzzling phenomenon. We trained two categories of Large-scale TS models, LLM4TS and TSFMs, across four scales, examining how architecture, model size, data volume, distribution, and training strategies influence model performance. Due to the lack of in-depth studies on representations in TS models, we examined the evolution of representations from both inter-layer and intra-layer perspectives. Our analysis reveals that only a small subset of layers play a critical role in learning, while the majority contribute minimally—a phenomenon we term **few-layer dominance**. Building on the insight, we propose a method to identify critical layers, allowing models to achieve performance on par while improving inference efficiency. Validation on 7 Large-scale TS models confirms the universality of few-layer dominance and the reliability of critical layers identification method. The code is available at anonymous-here.

## 1 INTRODUCTION

Time series (TS) forecasting is a critical task in many real-world applications, from financial market analysis to climate monitoring. Inspired by the success of large-scale models in other domains, the field has recently been dominated by two major paradigms: Time Series Foundation

Table 1: Performance degradation

| Metric: MAE | Sundial$_{small}$ | Sundial$_{Large}$ | Moirai$_{small}$ | Moirai$_{Large}$ | Time-LLM(G) | Time-LLM(L) |
|---|---|---|---|---|---|---|
| ETTh1 | 0.418 | 0.420 | 0.419 | 0.439 | 0.435 | 0.451 |
| ETTh2 | 0.387 | 0.387 | 0.337 | 0.343 | 0.398 | 0.400 |
| ETTm1 | 0.388 | 0.369 | 0.385 | 0.391 | 0.382 | 0.395 |
| ETTm2 | 0.324 | 0.315 | 0.337 | 0.343 | 0.332 | 0.325 |
| ECL | 0.265 | 0.262 | 0.274 | 0.270 | 0.261 | 0.270 |
| Weather | 0.271 | 0.275 | 0.281 | 0.273 | 0.268 | 0.265 |

Models (TSFMs) and Large Language Models for Time Series (LLM4TS) Xiaoming et al. (2025); Chang et al. (2025); Zhang et al. (2024); Woo et al. (2024). TSFMs are pretrained from scratch on massive TS datasets to learn universal temporal patterns. In contrast, LLM4TS models adapt powerful, text-trained LLMs to the time series domain through modality alignment. Despite their different approaches, both paradigms are driven by a shared belief inherited from other fields: that performance will monotonically improve with model scale.

However, emerging evidence challenge this foundational assumption Zhao et al. (2025); Liu et al. (2025c); Jin et al. (2023). In this paper, we investigate a counterintuitive phenomenon we term the *"scaling paradox"*: increasing model scale often fails to improve, and can even degrade, forecasting accuracy (as shown in Table 1, from Liu et al. (2025c); Pan et al. (2024)). This issue has remained under-explored, in part because its causes are complex. Performance is sensitive to a host of interacting factors—including architecture and model size, data volume and distribution, and training strategies. Furthermore, because time series data is itself a highly abstract, one-dimensional representation of the world, a deep understanding of its learned features is still largely lacking. Against this backdrop, our work investigates the relationship between internal representations and performance to offer clear insights into why scaling fails.

To systematically investigate this paradox, we construct and train distinct families of LLM4TS and TSFMs, each with four scales (Tiny, Small, Base, and Large). Our experiments confirm that the scaling paradox is a systemic issue, holding true across different model architectures and even when training data is substantially increased. This motivated a deeper analysis of the models' internal representations, which revealed a striking insight: in most large-scale TS models, *only a very small*

*subset of layers actively contributes* to learning temporal dynamics. We refer to this phenomenon as *"few-layer dominance."*

Based on this discovery, we propose a practical, model-agnostic method to identify and retain only these critical layers. This technique allows us to prune the redundant layers, resulting in significantly smaller and more efficient models that not only preserve but often improve upon the performance of the original, larger models. We validate the universality of our findings and the reliability of our method by applying it to 7 prominent, existing large-scale TS models, successfully demonstrating the widespread nature of few-layer dominance. Our main contributions are threefold:

1. We provide the first systematic evidence of a *scaling paradox* in large-scale time series models, rigorously demonstrating that "bigger is not always better" across both TSFM and LLM4TS paradigms.

2. We diagnose the root cause of this paradox, identifying **few-layer dominance**—where only a small subset of layers are functionally important—through inter-layer and intra-layer representation analysis.

3. We introduce and validate a practical **pruning method** that identifies and preserves these critical layers, yielding smaller, faster models that maintain or improve forecasting accuracy across a wide range of existing architectures.

## 2 RELATED WORK

### 2.1 LARGE-SCALE TIME SERIES MODELS

Large-scale time series (TS) models consist of two categories: LLM4TS and TSFMs. Applying pre-trained LLMs to TS forecasting poses a fundamental challenge: achieving modality alignment between text and TS(Liu et al., 2025b). Existing approaches either preserve the original LLM architecture or truncate some shallow layers to reduce computational complexity (Zhou et al., 2023; Liu et al., 2025b; Jin et al., 2023). The former can incur task-irrelevant overhead and semantic drift, while the latter weakens complete language capacity. TSFMs aim to build general-purpose TS models via learning from stratch on massive heterogeneous datasets, enabling cross-domain generalization (Wang et al., 2025; Goswami et al., 2024). They demonstrate remarkable potential in zero-shot scenarios and often introduce architecture modifications to capture intrinsic TS properties. However, most existing TSFMs either adopt similar architecture used for LLMs or make minor adjustments, overlooking the fundamental differences between TS and text in terms of semantic representations.

### 2.2 REPRESENTATION ANALYSIS IN LARGE-SCALE MODELS

Analyzing representations can reveal the roles of different components in semantic modeling. In the domain of language models, intermediate layers often encode more discriminative and robust features than the final layers, and transformer modules exhibit task-level "saturation events" during prediction, which can be modulated via intervention strategies to control the prediction trajectory (Csordás et al., 2025; Skean et al., 2025; Park et al., 2023). In multimodal large models, cross-modal representation analysis has shown that Multi-Head Self-Attention (MHSA) and Feed-Forward Network (FFN) capture distinct semantic features, with later layers primarily fine-tuning existing features rather than generating new computational patterns (Balasubramanian et al., 2024). In the TS domain, some studies have focused on semantic concepts learned by models and their positioning and manipulation in the latent space, enabling predictions to be guided without modifying model parameters (Wiliński et al., 2024). Nevertheless, systematic analyses of hidden representations and mechanisms in TS models remain scarce, limiting a deeper understanding.

## 3 BACKGROUND & PREPARATION

### 3.1 TIME SERIES FORECASTING TASK

Let $\boldsymbol{X} \in \mathbb{R}^{T \times V}$ denote the historical sequence of TS, where $T$ is sequence length and $V$ is variable dimension. The objective of TS forecasting is to learn a mapping function $\boldsymbol{F}_{\boldsymbol{\Theta}_{\boldsymbol{F}}} : \mathbb{R}^{T \times V} \to \mathbb{R}^{T' \times V}$, parameterized by $\boldsymbol{\Theta}_{\boldsymbol{F}}$, that predicts the future sequence over next $T'$ time steps. For additional explanations, please refer to Appendix B.1. Given a collection of $K$ distinct TS distributions $\{\mathcal{D}_1, \mathcal{D}_2, \ldots, \mathcal{D}_K\}$, can be formulated as:

$$\hat{\boldsymbol{X}}_{[T:T+T')} = \boldsymbol{F}_{\boldsymbol{\Theta}_{\boldsymbol{F}}}(\boldsymbol{X}_{[0:T)}), \boldsymbol{\Theta}_{\boldsymbol{F}}^* = \arg\min_{\boldsymbol{\Theta}_{\boldsymbol{F}}} \mathbb{E}_{k \sim \{1,\ldots,K\}} \mathbb{E}_{\boldsymbol{X}, \hat{\boldsymbol{X}} \sim \mathcal{D}_k} \left[ \mathcal{L}(\hat{\boldsymbol{X}}, \boldsymbol{X}_{[T:T+T')}) \right] \quad (1)$$

| Family | Scales | prior-para | layers | channels | learnable-para |
|--------|--------|-----------|--------|----------|----------------|
| | Tiny | GPT-2 | 6 | 768 | 3.92M |
| | Small | GPT-2 | 12 | 768 | 3.93M |
| LLM4TS | Base | Qwen-3 | 28 | 1024 | 0.16B |
| | Large | Qwen-3 | 28 | 2048 | 0.32B |
| | Tiny | RI | 6 | 768 | 85.02M |
| | Small | RI | 12 | 768 | 127.55M |
| LLM4TS | Base | RI | 28 | 1024 | 0.6B |
| | Large | RI | 28 | 2048 | 1.73B |

Figure 1: Architecture and specifications of Large-scale TS model family. (*Left*) We selected lightweight GPT-2 (Radford et al., 2018) and more powerful Qwen-3 (Yang et al., 2025) as backbones of LLM4TS, and TSFMs are trained end-to-end with full-parameter. (*Right*) A brief overview is provided here, with full details in E.1. "RI" denotes training from scratch with random initialization.

## 3.2 LARGE-SCALE TS MODELS ARCHITECTURE

LLM4TS are derived by fine-tuning of LLMs such as GPT (Radford et al., 2018) and LLaMA (Touvron et al., 2023), and TSFMs are pre-trained with random parameters. Although they vary in terms of model scale, architecture, and training strategies, they usually consist of three core components: time series embed, transformer blocks, and prediction head (Ansari et al., 2024; Liu et al., 2025c). More details are shown in Appendix in C.

**Time Series Embed.** (C.2) Patch-wise (Nie et al., 2023) and variable-wise (Liu et al., 2024b) embed are widely used to transform TS into embeddings. A raw sequence $\boldsymbol{X} \in \mathbb{R}^{T \times V}$, is encoded into $N$ tokens of dimension $d$, yielding the representation $\boldsymbol{E} \in \mathbb{R}^{N \times d}$.

**Transformer Blocks.** (C.1) After embedding, It is represented as $\boldsymbol{H}^{-1} \in \mathbb{R}^{N \times d}$. We denote $\boldsymbol{H}^l = \{\boldsymbol{h}_1^l, \boldsymbol{h}_2^l, \ldots, \boldsymbol{h}_N^l\} \in \mathbb{R}^{N \times d}$ as the output of the $l$-th layer ($0 \leq l \leq L-1$). Each block consists of a MHSA followed by a FFN, combined with residual connections and layer normalization (LN).

$$\boldsymbol{H}^l = \text{FFN}\big(\text{LN}(\boldsymbol{O}^l + \boldsymbol{H}^{l-1})\big) + \boldsymbol{O}^l + \boldsymbol{H}^{l-1}, \quad \boldsymbol{O}^l = \text{MHSA}\big(\text{LN}(\boldsymbol{H}^{l-1})\big) \tag{2}$$

$\boldsymbol{H}^{l-1} \in \mathbb{R}^{N \times d}$ are projected into queries, keys, and values: $\boldsymbol{Q}^{l,h} = \boldsymbol{H}^{l-1}\boldsymbol{W}_Q^{l,h}, \boldsymbol{K}^{l,h} = \boldsymbol{H}^{l-1}\boldsymbol{W}_K^{l,h}, \boldsymbol{V}^{l,h} = \boldsymbol{H}^{l-1}\boldsymbol{W}_V^{l,h}$ for each head $h \in \{1, \ldots, H\}$. $\boldsymbol{M}$ is the attention mask and $\boldsymbol{W}_O^l \in \mathbb{R}^{d \times d}$ is the output projection. Finally, outputs from all heads are concatenated and projected:

$$\boldsymbol{O}^l = \text{Concat}_{h=1}^H \big(\boldsymbol{A}^{l,h}\boldsymbol{V}^{l,h}\big)\boldsymbol{W}_O^l \tag{3}$$

**Prediction Head.** (C.3) $\boldsymbol{H}^{L-1}$ are mapped to the target space through a Prediction Head. $\boldsymbol{W}_{\text{out}} \in \mathbb{R}^{d \times D}$ is the learnable matrix, $D$ denotes the dimensionality of the prediction horizon, and $\boldsymbol{Y}$ represents the final forecasting sequence.

$$\boldsymbol{Y} = \text{LN}(\boldsymbol{H}^{L-1})\boldsymbol{W}_{\text{out}} \tag{4}$$

## 3.3 LARGE-SCALE TS MODELS FAMILY

When developing LLM4TS and TSFMs two families of TS models, We apply instance normalization and patching to mitigate semantic shifts from uneven data distributions and strengthen local information. TS embed and Prediction Head use linear probing, aligning hidden states with the dimensions of TS patches. Since variables are mostly independent, we adopt channel-independent strategy to reduce overfitting (Chen et al., 2025). For LLM4TS, only positional encoding and layer normalization are fine-tuned to adapt to downstream tasks and avoid catastrophic forgetting, whereas TSFMs are trained from scratch . See Figure 1, and more details are provided in Appendix C & E.1. Evaluation metrics are reported in Appendix B.2.

## 4 WHAT IMPACTS THE POTENTIAL OF LARGE-SCALE TS MODELS?

### 4.1 DOES BACKBONE SCALING IMPROVE FORECASTING PERFORMANCE? (RQ1)

Regardless of the model paradigm, the core of Large-scale TS models lies in a stack of Transformer Blocks, where MHSA and FFN capture local patterns and long-range dependencies (Zeng et al., 2023). Empirically, a more advanced backbone enhances performance, whether strength comes from powerful prior knowledge endowed by LLMs in LLM4TS, or from the fully learnable capacity of

Table 2: Statistics of datasets. Collected from multiple real-world scenarios.

| Datasets | ETTh1 | ETTh2 | ETTm1 | ETTm2 | Electricity | Exchange | Solar | Weather |
|---|---|---|---|---|---|---|---|---|
| Variates | 7 | 7 | 7 | 7 | 321 | 8 | 137 | 21 |
| Timesteps | 17,420 | 17,420 | 69,680 | 69,680 | 26,304 | 7,588 | 52,560 | 52,696 |
| Observations | 121,940 | 121,940 | 487,760 | 487,760 | 8,443,584 | 60,704 | 7,200,720 | 1,106,616 |

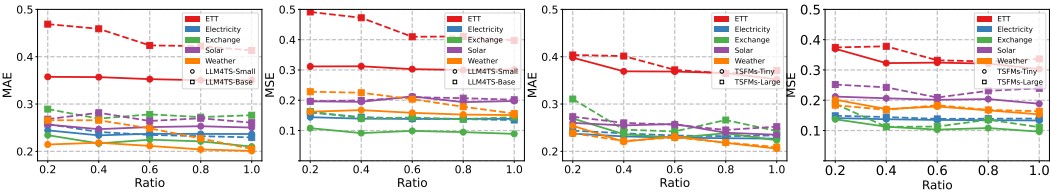

Figure 2: Performance of LLM4TS and TSFMs families across different backbones. (*Left*) Metrics of the LLM4TS family on 8 datasets. (*Right*) Metrics of the TSFMs family on 8 datasets. Lighter bars represent MAE, and darker bars represent MSE.

TSFMs trained from scratch (Isik et al., 2024; Zhuocheng et al., 2023; Chowdhery et al., 2022). However, in the field of TS, this theory does not seem to hold. We examine the performance of Large-scale models across different datasets in Table 2, focusing on whether backbone scaling brings substantial gains. We follow FSCA (Hu et al., 2025) for dataset partitioning, using a input length of 336 and a prediction length of 96. More datasets details are provided in Appendix D. Detailed experimental settings are provided in Appendix E.2. Figure 2 shows MAE and MSE do not significant decrease as the backbone scales up. Interestingly, most datasets reveal an anomalous trend, with larger backbones showing degraded performance. Full results are reported in Appnedix G.1.

> **Takeaways:** In the single-dataset setting, backbone scaling does not improve performance, revealing a clear gap between backbone and performance. Neither the prior knowledge in LLM4TS nor the stronger parameterization in TSFMs directly translates into learning TS representation better.

## 4.2 DOES LIMITED DATA VOLUME HINDER THE ADVANTAGES OF BACKBONE? (RQ2)

Figure 3: Performance under training data ratios from 20% to 100%. (*Left two*) Performance of LLM4TS family at Small and Base scales. (*Right two*) Performance of TSFMs family at the Tiny and Large scales.

When data volume increases, MAE and MSE exhibit an downward trend, enabling models to learn more generalizable representations, shown in Figure 3. However, models with more advanced backbone still underperform compared to weaker counterparts, and more training data cannot fully unlock model's potential. Detailed experimental settings are provided in Appendix E.2. Full results are reported in Appnedix G.2.

> **Takeaways:** Data volume is not the primary factor limiting Large-scale TS models to leverage their scale. Although increasing the ratio allows models to learn more effective representations, advanced backbones still do not consistently outperform.

## 4.3 DOES PERFORMANCE DEGRADATION STEM FROM DATASET HOMOGENEITY? (RQ3)

Training on a narrow data source, such as a single energy dataset, can cause the model to overfit to specific temporal resolutions and limited semantic patterns (Huang et al., 2025; Woo et al., 2024). Such constraints are particularly detrimental for Large-scale TS Models with substantial trainable parameters, as they fail to fully activate model potential. We apply a cross-dataset learning strategy,

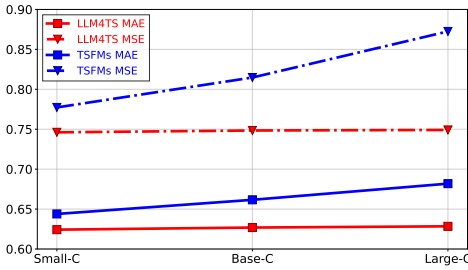

Figure 4: Dataset sources

Figure 5: Out-of-domain performance

Table 3: Performance under single-dataset and cross-dataset learning. Red and blue mark best performance of LLM4TS, TSFMs. (*Left*) Single-dataset learning, (*Right*) Cross-dataset learning.

| | LLM4TS | | | TSFMs | | |
|---|---|---|---|---|---|---|
| | Small | Base | Large | Small | Base | Large |
| **MAE** | 0.2875 | 0.3250 | 0.3283 | 0.2921 | 0.3028 | 0.3078 |
| **MSE** | 0.2226 | 0.2784 | 0.2846 | 0.2286 | 0.2482 | 0.2540 |

| | LLM4TS | | | TSFMs | | |
|---|---|---|---|---|---|---|
| | Small-C | Base-C | Large-C | Small-C | Base-C | Large-C |
| **MAE** | 0.2986 | 0.2894 | 0.2875 | 0.2872 | 0.2986 | 0.3006 |
| **MSE** | 0.2367 | 0.2263 | 0.2233 | 0.2257 | 0.2496 | 0.2543 |

training on the combined training sets and sharing model weights across test sets from all datasets. TS data are collected from diverse domains, as illustrated in Figure 4. Other settings are identical to subsection 4.1 and 4.2. Main results are reported in Table 3, where for fair comparison, MAE & MSE is averaged over in-domain performance across datasets. Detailed experimental settings are provided in Appendix E.3. Full results are provided in Appendix G.3.

For LLM4TS family, cross-dataset learning improves in-domain performance but remains inferior to LLM4TS-Small. Cross-dataset learning partly activates the backbone's ability to leverage LLM knowledge, but LLM4TS with Qwen-3 as the backbone underperforms GPT-2, implying that stronger priors do not guarantee extra gains. For the TSFMs family, diverse training distributions enhance representation learning and improve forecasting performance, but larger model scales unfortunately lead to performance degradation. Beyond in-domain evaluation, we assess out-of-domain performance by testing on datasets unseen during training (Goswami et al., 2024). As shown in Figure 5, model performance exhibits a clear degradation trend. It is evaluated on eight datasets, including NN5, PDB, Sceaux, Smart, Spanish, Sunspot Rain, US Births, and Wind Power, with details provided in Appendix D.2. In summary, cross-dataset learning significantly enhances the representation diversity of Large-scale TS models, but benefit does not improve with model size, as larger parameter counts fail to bring forecasting gains.

> **Takeaways:** For both LLM4TS and TSFMs families, enhancing training data diversity does not effectively translate into performance gains, either in-domain or out-of-domain tasks.

### 4.4 DO ALL LAYERS CONTRIBUTE TO FINAL PREDICTIONS? (RQ4)

Since the summaries in subsection 4.2 and 4.3 reveal that neither data volume nor diversity of data distribution is the primary factor limiting the gains from scaling, We put forward an intriguing hypothesis: do all backbone layers actively support the final predictions, or does deeper stacking induce layer inertia? Therefore, We investigate layers by analyzing **inter-layer** and **intra-layer representations** to understand how embeddings evolve throughout the model. The former treats each Transformer layer as a whole, and the latter delves into the internal components. Representations undergo shifts in both magnitude and directional aspects when passing through different modules. Magnitude shifts reflect scaling effects, and directional shifts capture migration within the semantic subspace, together providing a principled measure (Wiliński et al., 2024).

**Inter-layer representations**. We use Euclidean distance to measure absolute vector differences and reflects overall scaling variations in a geometrical way. Let $\boldsymbol{H}^{l-1}$ and $\boldsymbol{H}^l$ denote the input and output of $l$-th layer. Directional shifts are measured using similarity metrics like Cosine Similarity and CKA similarity (Kornblith et al., 2019), with Cosine Similarity being the most common due to its scale invariance. The directional shifts are presented below, and $\langle \cdot, \cdot \rangle$ represents the inner product.

$$Dist^l = \|\boldsymbol{H}^l - \boldsymbol{H}^{l-1}\|_2, \quad Sim(\boldsymbol{H}^l, \boldsymbol{H}^{l-1}) = \frac{\langle \boldsymbol{H}^l, \boldsymbol{H}^{l-1} \rangle}{\|\boldsymbol{H}^l\|_2 \|\boldsymbol{H}^{l-1}\|_2} \quad (5)$$

**Intra-layer representations.** Each head in the MHSA module can be regarded as an independent relational learner (Liu et al., 2021), parameterized by distinct projection matrices. Given the embeddings

$\boldsymbol{X}^l \in \mathbb{R}^{N \times d_{\text{model}}}$ at layer $l$, the $i$-th head generates its attention weights:

$$\boldsymbol{A}_i^l = \text{softmax}\left(\boldsymbol{Q}_i^l \boldsymbol{K}_i^{l\top}/\sqrt{d}\right), \quad \boldsymbol{Q}_i^l = \boldsymbol{X}^l \boldsymbol{W}_i^Q, \ \boldsymbol{K}_i^l = \boldsymbol{X}^l \boldsymbol{W}_i^K \tag{6}$$

$\boldsymbol{W}_i^Q, \boldsymbol{W}_i^K \in \mathbb{R}^{d_{\text{model}} \times d}$ are head-specific projections and $d = d_{\text{model}}/H$ denotes the per-head dimensionality. We measure the average pairwise similarity across all head attentions in layer $l$. A lower $\bar{s}^{(l)}$ indicates higher functional diversity among heads, reflecting that each head captures distinct aspects of the TS representations. $\bar{s}^l$ be defined as:

$$\bar{s}^l = \frac{2}{H(H-1)} \sum_{1 \le i < j \le H} \text{sim}\left(\boldsymbol{A}_i^l, \boldsymbol{A}_j^l\right) \tag{7}$$

Figure 6 shows LLM4TS and TSFMs families of different scales, only few-layer contribute substantial representation shifts, while many layers function in a redundant manner rather than actively driving learning. This layer-wise representation phenomenon also appears in cross-dataset learning models and those trained with varying data sizes, and more details are provided in Appendix XXX. Two

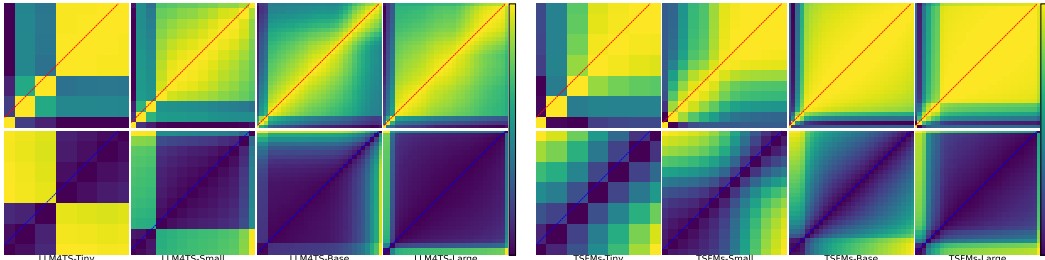

LLM4TS-Tiny    LLM4TS-Small    LLM4TS-Base    LLM4TS-Large      TSFMs-Tiny    TSFMs-Small    TSFMs-Base    TSFMs-Large

Figure 6: Inter-layer cosine similarity (*row 1*) and Euclidean distance (*row 2*). Brighter areas indicate higher values, darker areas lower values. It can be observed that across eight models of different scales and architectures, inter-layer representations exhibit high similarity and low Euclidean distance.

families exhibit similar trends in intra-layer representations. For models with smaller backbones (6 or 12 layers), heads in the shallow layers primarily allocate their attention to distinct semantic patterns. However, in the middle and deeper layers, $\bar{s}^l$ approaches 1, with most heads learning within the similar subspace. For models with larger backbones (28 layers), heads across all layers almost display such learning characteristics, exhibiting high similarity, shown in Figure 7.

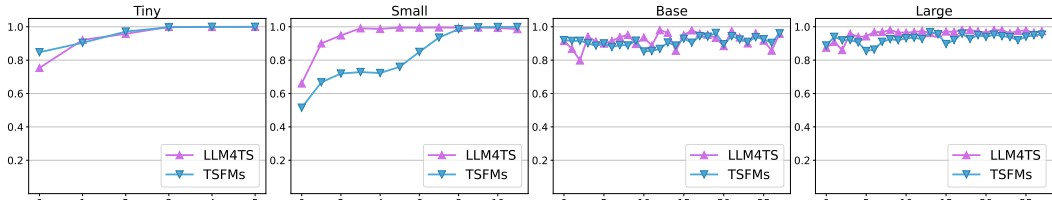

Figure 7: Average inter-layer pairwise similarity across all head attentions.

> **Takeaways:** For LLM4TS and TSFMs families, only a small fraction of layers actively contribute to representation learning, and the vast majority remain largely passive spectators.

## 4.5 CAN A FEW LAYERS MATCH FULL-MODEL FORECASTING PERFORMANCE? (RQ5)

Building on the above, different layers vary greatly in their capacity to transform representations, with only a small subset making the primary contribution. Then, if only a few seemingly more influential layers are retained while pruning the remaining "less important" ones, would the model still exhibit comparable performance? This motivates the need for a method to accurately assess each layer's contribution and identify critical layers. Due to the progressive process, successive layer re-integrates and refines the representations produced by preceding layers. Earlier layers affect subsequent layers, with their influence diminishing progressively. Therefore, static properties of an individual layer may not fully reflect its contribution to last predictions. Considering the decaying influence of preceding layers on deeper layers, with formula 7, importance score of the $l$-th layer is:

$$I^l = \mathbf{1}\{l \in \text{Top-}\tau\%\} \cdot \left(1 - R^l\right) \cdot \left(1 - \bar{s}^l\right), \quad R^l = \sum_{k=1}^{K} w_k \cdot Sim\left(\boldsymbol{H}^l, \boldsymbol{H}^{l-k}\right) \tag{8}$$

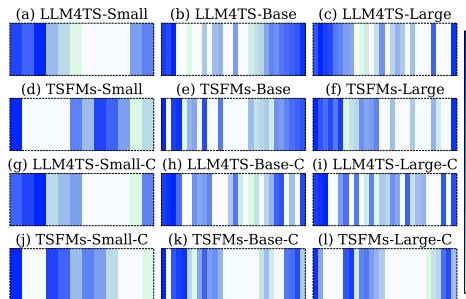

(a) LLM4TS-Small   (b) LLM4TS-Base   (c) LLM4TS-Large

(d) TSFMs-Small   (e) TSFMs-Base   (f) TSFMs-Large

(g) LLM4TS-Small-C   (h) LLM4TS-Base-C   (i) LLM4TS-Large-C

(j) TSFMs-Small-C   (k) TSFMs-Base-C   (l) TSFMs-Large-C

Figure 8: Average layer-wise importance

Table 4: Critical layer identification

**Algorithm: Critical Layer Identification**
**Input:** Representations $\boldsymbol{H}^0, \ldots, \boldsymbol{H}^{L-1}$, attention weights $\boldsymbol{A}_i^l$,
   decay factor $\alpha$, top percentile $\tau$, number of preceding layers $K$
**Output:** Ordered list of critical layers
**for** $l = 0$ **to** $L - 1$ **do**
   $Dist^l \leftarrow \|\boldsymbol{H}^l - \boldsymbol{H}^{l-1}\|_2, \leftarrow Sim(\boldsymbol{H}^l, \boldsymbol{H}^{l-1})$
   $\bar{s}^l \leftarrow \frac{1}{\binom{H}{2}} \sum_{i<j} Sim(\boldsymbol{A}_i^l, \boldsymbol{A}_j^l)$
   $w_k \leftarrow \alpha^k / \sum_{i=1}^K \alpha^i, \quad R^l \leftarrow \sum_{k=1}^K w_k \cdot Sim(\boldsymbol{H}^l, \boldsymbol{H}^{l-k})$
   $I^l \leftarrow (1 - R^l) \cdot (1 - \bar{s}^l)$
**end for**
**Let** $\mathcal{S}$ be the indices of Top-$\tau$% layers ranked by $Dist^l$.
**for** $l = 1$ **to** $L - 2$ **do**
   **if** $l \notin \mathcal{S}$ **then** $I^l \leftarrow 0$
**end for**
**Rank** all layers by $I^l$ in descending order:

$\alpha \in (0, 1)$ is a decay factor, $w_k = \frac{\alpha^k}{\sum_{i=1}^K \alpha^i}$ is the normalized decaying weight, and $K$ denotes the number of preceding layers considered. It reflects a joint requirement that a layer is important only when it has low inter-layer representational similarity and high intra-layer head attention diversity. $\tau$ specifies the top percentile of layers retained. By ranking all layers according to $I^l$ in descending order, those contributing most can be identified. The first and last layers connect TS and embeddings and play a critical role in semantic transformation, so they are exempt from the Top-$\tau$% selection. Details are summarized in Table 4.

Due to space limitations, Figure 8 presents average distribution of layer importance across validation sets of different datasets, where darker colors indicate greater importance. Based on importance ranking, we select a subset of layers and prune the rest. The pruned model is fine-tuned on training sets to re-align with data distribution, and main results are reported in Table 5. More results and details are reported in Appendix G.4.

Table 5: Prediction performance and inference efficiency under pruning. All experiments are conducted under identical settings and hyperparameters. Efficiency is measured as the ratio of the original model's inference time to that of the pruned model. "–" means training and testing on a single dataset, without considering out-of-domain performance.

| Model | avg MAE-ID | | avg MSE-ID | | avg MAE-OOD | | avg MSE-OOD | | Efficiency↑ |
|---|---|---|---|---|---|---|---|---|---|
| | Pruned | Original | Pruned | Original | Pruned | Original | Pruned | Original | |
| LLM4TS-Small | 0.283 | 0.288 | 0.218 | 0.223 | – | – | – | – | 2.68 |
| LLM4TS-Base | 0.293 | 0.325 | 0.226 | 0.278 | – | – | – | – | 2.21 |
| LLM4TS-Large | 0.283 | 0.328 | 0.219 | 0.285 | – | – | – | – | 2.37 |
| TSFMs-Small | 0.287 | 0.292 | 0.221 | 0.227 | – | – | – | – | 2.45 |
| TSFMs-Base | 0.284 | 0.303 | 0.220 | 0.248 | – | – | – | – | 2.74 |
| TSFMs-Large | 0.290 | 0.308 | 0.231 | 0.254 | – | – | – | – | 2.69 |
| LLM4TS-Small-C | 0.298 | 0.299 | 0.233 | 0.237 | 0.625 | 0.624 | 0.740 | 0.746 | 2.16 |
| LLM4TS-Base-C | 0.288 | 0.289 | 0.225 | 0.226 | 0.618 | 0.627 | 0.738 | 0.748 | 2.09 |
| LLM4TS-Large-C | 0.286 | 0.288 | 0.226 | 0.223 | 0.620 | 0.628 | 0.727 | 0.749 | 2.16 |
| TSFMs-Small-C | 0.286 | 0.287 | 0.227 | 0.226 | 0.635 | 0.644 | 0.782 | 0.777 | 1.88 |
| TSFMs-Base-C | 0.293 | 0.299 | 0.247 | 0.250 | 0.654 | 0.662 | 0.773 | 0.815 | 2.49 |
| TSFMs-Large-C | 0.282 | 0.301 | 0.224 | 0.254 | 0.659 | 0.682 | 0.821 | 0.872 | 2.65 |

> **Takeaways:** Retaining only critical layers with fine-tuning preserves or even improves forecasting accuracy, while substantially reducing parameters and inference latency.

## 5 EXPERIMENTS & ANALYSIS

### 5.1 METHOD ADAPTATION FOR BASELINES

In addition to the datasets presented in Table 2, we incorporate the Traffic as well as four subsets of PEMS. We analyze 4 LLMsTS (Hu et al., 2025; Liu et al., 2025b; Jin et al., 2023; Zhou et al., 2023) and 3 TSFMs (Liu et al., 2025c; Ansari et al., 2024; Das et al., 2024)well-known works to validate the transferability of our approach. They differ in architecture, encoder–decoder & decoder-only paradigms, choices of TS Embed and Prediction Head, and scales.

First, we conduct an analysis of both inter-layer and intra-layer representations on the validation set. Following the procedure introduced earlier, we compute the cosine similarity between the input and output representations across layers, as well as the pairwise similarity of attention weights

among heads within each layer, to derive a score for every layer. In addition, except for the first and last layers, we incorporate the Euclidean distance between inputs and outputs to characterize the magnitude shifts of inter-layer transformations. If the Euclidean distance difference of a given layer does not fall within the top 80% among all layers, its score is set to zero and it is excluded from the ranking of important layers, shown in Table 4.

Based on this criterion, we obtain a hierarchical importance ranking of layers. We then select the top 50% layers according to this ranking for retention. Finally, adhering to the original training strategy of baseline, we fine-tune the pruned architecture on training sets to re-align model with the underlying data distribution.

Table 6: Method adaptation for baselines. Multivariate inputs bring no significant gains, as all time series are fed in a channel-independent manner. Input length, prediction length, backbone, and hyperparameter settings are identical to baselines. Main results reported below are averaged across different horizons, experiment settings and full results are provided in Appendix G.5.

| Models | FSCA | | | | CALF | | | | Time-LLM(G) | | | | OFA | | | | Sundial | | | | Chronos | | | | TimesFM | | | |
|---|---|---|---|---|---|---|---|---|---|---|---|---|---|---|---|---|---|---|---|---|---|---|---|---|---|---|---|---|
| | Pruned | | Original | | Pruned | | Original | | Pruned | | Original | | Pruned | | Original | | Pruned | | Original | | Pruned | | Original | | Pruned | | Original | |
| Metric | MAE | MSE | MAE | MSE | MAE | MSE | MAE | MSE | MAE | MSE | MAE | MSE | MAE | MSE | MAE | MSE | MAE | MSE | MAE | MSE | MAE | MSE | MAE | MSE | MAE | MSE | MAE | MSE |
| ETTh1 | 0.443 | 0.426 | 0.444 | 0.430 | 0.431 | 0.436 | 0.434 | 0.446 | 0.446 | 0.431 | 0.459 | 0.448 | 0.435 | 0.429 | 0.434 | 0.430 | 0.504 | 0.534 | 0.517 | 0.556 | 0.395 | 0.470 | 0.428 | 0.494 | 0.471 | 0.430 | 0.485 | 0.449 |
| ETTh2 | 0.390 | 0.349 | 0.390 | 0.348 | 0.391 | 0.362 | 0.395 | 0.373 | 0.408 | 0.368 | 0.410 | 0.370 | 0.403 | 0.366 | 0.403 | 0.366 | 0.431 | 0.417 | 0.439 | 0.424 | 0.306 | 0.261 | 0.320 | 0.270 | 0.395 | 0.408 | 0.411 | 0.415 |
| ETTm1 | 0.385 | 0.350 | 0.387 | 0.352 | 0.384 | 0.387 | 0.387 | 0.391 | 0.390 | 0.358 | 0.389 | 0.359 | 0.388 | 0.357 | 0.385 | 0.355 | 0.405 | 0.381 | 0.425 | 0.415 | 0.516 | 0.664 | 0.518 | 0.677 | 0.414 | 0.391 | 0.429 | 0.408 |
| ETTm2 | 0.319 | 0.258 | 0.320 | 0.259 | 0.320 | 0.268 | 0.318 | 0.277 | 0.327 | 0.268 | 0.330 | 0.274 | 0.325 | 0.265 | 0.325 | 0.265 | 0.353 | 0.310 | 0.359 | 0.313 | 0.277 | 0.202 | 0.286 | 0.207 | 0.356 | 0.366 | 0.339 | 0.357 |
| Electricity | 0.266 | 0.165 | 0.267 | 0.166 | 0.269 | 0.184 | 0.272 | 0.189 | 0.263 | 0.163 | 0.271 | 0.169 | 0.259 | 0.166 | 0.265 | 0.169 | 0.261 | 0.162 | 0.263 | 0.162 | 0.315 | 0.262 | 0.319 | 0.264 | 0.141 | 0.223 | 0.157 | 0.232 |
| Exchange | 0.434 | 0.436 | 0.449 | 0.450 | 0.413 | 0.378 | 0.408 | 0.370 | 0.448 | 0.436 | 0.448 | 0.441 | 0.408 | 0.384 | 0.428 | 0.410 | 0.437 | 0.418 | 0.461 | 0.473 | 0.158 | 0.063 | 0.161 | 0.061 | 0.185 | 0.089 | 0.179 | 0.092 |
| Soalr | 0.264 | 0.198 | 0.274 | 0.210 | 0.260 | 0.227 | 0.268 | 0.251 | 0.264 | 0.190 | 0.257 | 0.189 | 0.264 | 0.211 | 0.276 | 0.216 | 0.255 | 0.195 | 0.262 | 0.198 | 0.503 | 0.566 | 0.512 | 0.587 | | | | |
| Traffic | 0.277 | 0.390 | 0.284 | 0.397 | 0.284 | 0.424 | 0.286 | 0.465 | 0.279 | 0.395 | 0.284 | 0.398 | 0.275 | 0.404 | 0.298 | 0.422 | 0.285 | 0.414 | 0.302 | 0.424 | 0.419 | 0.641 | 0.428 | 0.671 | 0.445 | 0.463 | 0.434 | 0.479 |
| Weather | 0.269 | 0.233 | 0.268 | 0.229 | 0.271 | 0.255 | 0.277 | 0.259 | 0.266 | 0.228 | 0.266 | 0.228 | 0.257 | 0.230 | 0.268 | 0.232 | 0.276 | 0.242 | 0.279 | 0.246 | 0.189 | 0.174 | 0.191 | 0.172 | 0.216 | 0.209 | 0.231 | 0.255 |
| PEMS03 | 0.264 | 0.171 | 0.270 | 0.171 | 0.401 | 0.363 | 0.406 | 0.373 | 0.282 | 0.183 | 0.284 | 0.185 | 0.268 | 0.178 | 0.281 | 0.183 | 0.258 | 0.159 | 0.264 | 0.168 | 0.526 | 0.534 | 0.542 | 0.561 | 0.496 | 0.544 | 0.524 | 0.571 |
| PEMS04 | 0.346 | 0.458 | 0.358 | 0.463 | 0.508 | 0.772 | 0.513 | 0.778 | 0.376 | 0.482 | 0.379 | 0.484 | 0.362 | 0.477 | 0.375 | 0.488 | 0.352 | 0.454 | 0.360 | 0.463 | 0.526 | 0.589 | 0.552 | 0.602 | 0.544 | 0.598 | 0.537 | 0.590 |
| PEMS07 | 0.221 | 0.115 | 0.242 | 0.130 | 0.394 | 0.384 | 0.405 | 0.393 | 0.250 | 0.137 | 0.253 | 0.142 | 0.235 | 0.136 | 0.252 | 0.145 | 0.233 | 0.134 | 0.218 | 0.122 | 0.546 | 0.643 | 0.535 | 0.643 | 0.542 | 0.629 | 0.576 | 0.651 |
| PEMS08 | 0.345 | 0.510 | 0.353 | 0.509 | 0.488 | 0.770 | 0.506 | 0.837 | 0.377 | 0.527 | 0.386 | 0.544 | 0.359 | 0.510 | 0.387 | 0.557 | 0.356 | 0.511 | 0.359 | 0.520 | 0.579 | 0.774 | 0.586 | 0.782 | 0.576 | 0.722 | 0.593 | 0.764 |

## 5.2 FULL VS. PRUNED

FSCA, OFA and CALF adopt a coarse-grained truncation strategy of only few layers of LLM. In this subsection, we explore performance under the full-LLM setting on ETT and Weather datasets. **Pruned** refers to models built upon full LLM but retaining some critical layers followed by fine-tuning. **W** replace incomplete LLM (GPT-2 with 6 layers) in baseline with full LLM (GPT-2 with 12 layers), and **R** randomly keep same number of layers . Other settings are the same as in subsection 5.1. Table 7 shows pruned models not only outperform the incomplete backbone of Table 6, but also surpass the models using full backbone. Randomly selecting same number of layers yields suboptimal performance, meaning contribution of each layer is far from arbitrary.

Table 7: Performance of different-layer backbones

| Models | FSCA | | | | | | CALF | | | | | | OFA | | | | | |
|---|---|---|---|---|---|---|---|---|---|---|---|---|---|---|---|---|---|---|
| | Pruned | | W | | R | | Pruned | | W | | R | | Pruned | | W | | R | |
| Metric | MAE | MSE | MAE | MSE | MAE | MSE | MAE | MSE | MAE | MSE | MAE | MSE | MAE | MSE | MAE | MSE | MAE | MSE |
| ETTh1 | 0.443 | 0.426 | 0.457 | 0.442 | 0.447 | 0.432 | 0.431 | 0.436 | 0.440 | 0.445 | 0.452 | 0.461 | 0.435 | 0.429 | 0.451 | 0.461 | 0.446 | 0.451 |
| ETTh2 | 0.390 | 0.349 | 0.402 | 0.367 | 0.398 | 0.355 | 0.391 | 0.362 | 0.396 | 0.374 | 0.394 | 0.374 | 0.403 | 0.366 | 0.424 | 0.379 | 0.418 | 0.384 |
| ETTm1 | 0.385 | 0.350 | 0.399 | 0.368 | 0.394 | 0.360 | 0.384 | 0.387 | 0.395 | 0.392 | 0.402 | 0.397 | 0.388 | 0.357 | 0.397 | 0.365 | 0.395 | 0.366 |
| ETTm2 | 0.319 | 0.258 | 0.324 | 0.267 | 0.316 | 0.258 | 0.320 | 0.268 | 0.326 | 0.280 | 0.325 | 0.279 | 0.325 | 0.265 | 0.337 | 0.287 | 0.335 | 0.282 |
| Weather | 0.269 | 0.233 | 0.265 | 0.237 | 0.267 | 0.244 | 0.271 | 0.255 | 0.279 | 0.267 | 0.283 | 0.278 | 0.257 | 0.230 | 0.268 | 0.248 | 0.267 | 0.231 |

## 5.3 INFERENCE OVERHEAD AND PERFORMANCE

We retain different numbers of layers in descending order of importance to explore the inference efficiency and performance of SUNDIAL and TIME-LLM, and metrics are averaged over horizons ∈ {96, 192, 336, 720}. Inference overhead is defined as the ratio between the pruned model's inference time and that of the unpruned ones under identical conditions. Figure 9 shows a few layers contribute almost all performance, yet removing entire backbone causes a clear drop in capability.

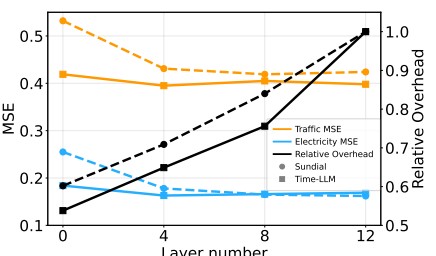
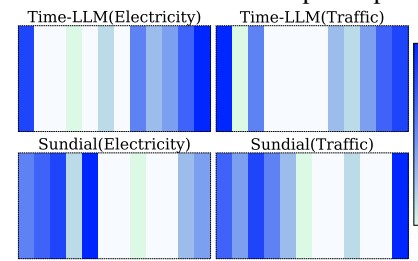

Figure 9: Overhead and MSE

Table 8: Different strategies

| Dataset | OFA | | | | Sundial | | |
|---|---|---|---|---|---|---|---|
| | original | vanilla | w/o | full-param | original | vanilla | w/o |
| **ETTh1** | | | | | | | |
| MAE | 0.434 | 0.435 | 0.554 | 0.472 | 0.556 | 0.534 | 0.573 |
| MSE | 0.430 | 0.429 | 0.627 | 0.486 | 0.517 | 0.504 | 0.670 |
| **Exchange** | | | | | | | |
| MAE | 0.428 | 0.408 | 0.495 | 0.449 | 0.461 | 0.437 | 0.475 |
| MSE | 0.410 | 0.384 | 0.481 | 0.463 | 0.473 | 0.418 | 0.501 |

Table 9: Results for classification task

| Dataset | FSCA | | Time-LLM(G) | | OFA | |
|---|---|---|---|---|---|---|
| | pruned | original | pruned | original | pruned | original |
| Handwriting | 38.8 | 38.6 | 33.4 | 33.6 | 32.9 | 32.7 |
| Heartbeat | 79.1 | 78.7 | 78.4 | 78.6 | 77.2 | 77.2 |
| JapaneseVowels | 98.6 | 98.9 | 98.3 | 97.8 | 98.7 | 98.5 |

## 6  ABLATIONS

**Re-alignment Strategy.**  We examine the effect of re-alignment strategies on {**LLMsTS** : OFA, **TSFMs** : SUNDIAL}. Original denotes baseline without pruning; **vanilla** refers to fine-tuning after retaining a few layers; **w/o** means infer after removing layers; and full-para means full-parameter fine-tuning. Metrics are averaged over horizons $\in \{96, 192, 336, 720\}$. Table 8 shows align-free method fails to bridge gap between structure and parameters distribution, adopting full-parameter may incur catastrophic forgetting (Kirkpatrick et al., 2017).

**Classification Task.** It reveals potential in classification tasks, and a small number of important layers are sufficient to achieve comparable or better performance, shown in Table 9.

## 7  DISCUSSION AND CONCLUSION

### 7.1  FROM REPRESENTATION TO PREDICTION: A PROGRESSIVE PERSPECTIVE

The hidden states from each layer are sequentially fed into Prediction Head to generate 96-step forecasts. Notably, under a train-free pruning scheme that preserves only first and last layers (Cols. 3 & 6), model still delivers competitive performance. In contrast, directly projecting intermediate-layer representations back to the TS yields outputs that diverge from ground-truth distribution (Cols. 2 & 5). For reference, (Cols. 1 & 4) correspond to the full model, where Prediction Head consumes representations from the initial state (input embedding with positional encoding only) and final layer.

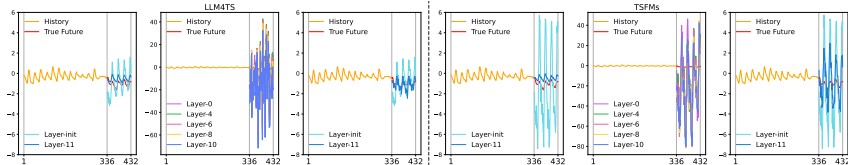

Figure 10: Projecting hidden states from different layers into the Prediction Head. (H.3)

### 7.2  ATTENTION WEIGHTS AS A WINDOW INTO PATTERN LEARNING FOR TS

Attention entropy $\mathcal{H}^l$ measures the degree of dispersion in how each head allocates its attention across tokens(Zhang et al., 2025). Interestingly, $\mathcal{H}^l$ remains stable across layers, meaning limited change in attention dispersion. Moreover, heads often concentrate on specific tokens, with a few positions—especially last one, exerting dominant influence on prediction, as key subsequences near target typically provide most salient cues. If very few tokens determine the prediction, it is understandable that few layers still perform well (Wen et al., 2025; Kim et al., 2022), and a small set of informative token features may suffice without extensive parameters. (H.3)

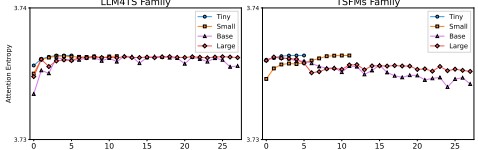 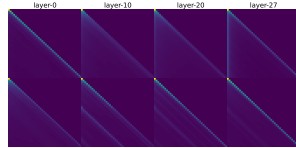

Figure 11: (*Left two*) $\mathcal{H}^l$ of models with different scales; (*Upper-right*) attention weights of LLM4TS-Base, (*Lower-right*) attention weights of TSFMs-Base.

### 7.3  CONTRIBUTION

In this work, we conduct evaluations of Large-scale TS models across architectures, scales, data, distributions, and training strategies, provide the first in-depth analysis of inter- and intra-layer representations, reveal the phenomenon of few-layer dominance whereby only a small subset of layers are critical, and further propose a critical-layer identification method that preserves forecasting accuracy while improving inference efficiency, with extensive validation on seven representative models confirming its universality and effectiveness.

# 8 THE USE OF LARGE LANGUAGE MODELS

LLMs were not involved in the design, execution, or analysis of this research. Their usage was limited to language polishing and writing assistance. We are solely responsible for the authenticity, accuracy, and validity of the research presented in our work.

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

# A  MORE RELATED WORK

## A.1  LARGE LANGUAGE MODELS FOR TIME SERIES

Applying fully pre-trained LLMs to TS tasks poses the central challenge of achieving effective modality alignment . Existing efforts can be broadly categorized into two directions: TS embedding and latent space alignment (Liu et al., 2025b; Pan et al., 2024). TS embedding maps raw TS into representations that are more compatible with the pre-trained LLMs space, simplifying knowledge transfer (Liu et al., 2024a). When combined with prompt-based mechanisms, the strategy leverages intrinsic capacity of LLMs to process TS tasks in a form that preserves generalization abilities (Liu et al., 2025a). By contrast, latent-space alignment seeks to directly bridge representational gap. It is typically achieved through joint training, contrastive learning, or other optimization strategies that enforce deeper cross-modal alignment, enhancing feature sharing and transferability across modalities. One line of work retains original structure of LLMs, mapping TS to input layer (Jin et al., 2023). It maximizes preservation of pretrained parameters and prior knowledge learned from corpora, avoiding instability and retraining overhead that structural alterations may incur. However, such direct adoption often overlooks issues of parameter compatibility, which may compromise efficiency and adaptability in applications. In contrast, another line of work employs only a few shallow layers, which reduces computational complexity and parameter scale, improve inference efficiency and alleviate deployment costs. However, it's drawback lies in the loss of pretrained knowledge, diminishing ability to capture local patterns and long-range dependencies.

## A.2  TIME SERIES FOUNDATION MODELS

TSFMs aim to construct large-scale, general-purpose models for TS tasks by learning on massive heterogeneous datasets, enabling generalized capabilities across tasks (Ansari et al., 2024; Das et al., 2024; Liu et al., 2025c). TSFMs have demonstrated remarkable potential in zero-shot scenarios, where they can deliver accurate predictions on unseen domains, variable types and temporal granularities, reducing the reliance on task-specific training data that traditional methods typically require. To better accommodate inherent properties of TS—such as continuity, trend, and multi-scale dynamics (Liu et al., 2024c), TSFMs typically incorporate customized adaptations of Transformer architecture. At the same time, they rely on large-scale, cross-domain pretraining to capture diverse patterns from multi-domains. However, most current TSFMs either directly adopt or only slightly modify LLMs architectures, overlooking fundamental discrepancies between TS and text in terms of statistical properties, dependency structures, and semantic representations. Consequently, designing architectures and training strategies that explicitly account for statistical and structural uniqueness of TS remains an open challenge for this field. Although current TSFMs have achieved notable performance, they still exhibit limitations in parameter and efficiency, deployment feasibility, inference latency, and cross-task adaptability.

## A.3  TIME SERIES REPRESENTATION LEARNING

Representation learning is an indispensable research direction in TS analysis. Effective representations can capture important patterns of TS, reduce redundancy, and enhance performance of downstream tasks. Among current approaches, multi-scale pyramid structures decompose the original TS into hierarchical features at different resolutions, reducing computational overhead (Liu et al., 2022). Frequency-domain decomposition and frequency-enhanced attention mechanisms, leveraging Fourier or wavelet transforms, optimize seasonal pattern modeling and enable low-complexity representation learning (Zhou et al., 2022). TS can thus be decomposed into seasonal, trend, and residual components to capture multi-scale dynamic patterns (Wu et al., 2021). Next-token prediction employ patch-based inputs, discretization, or modality alignment to pre-train models, enhancing numerical reasoning and inference capabilities (Cao et al., 2023; Xue & Salim, 2023). Multi-task learning further integrates reconstruction and prediction errors to improve robustness and expressiveness of learned representations (Liu et al., 2024a). Despite these advances, investigations into hidden representations and underlying mechanisms of TS models remain limited, which constrains a deeper understanding of their semantic modeling capacity and reasoning process, and affects interpretability and transferability in complex downstream tasks.

# B  PRELIMINARY

## B.1  PROBLEM DEFINITION

TS forecasting constitutes one of the most challenging tasks in TS analysis, due to the inherent non-stationarity, complex temporal dependencies, potential presence of exogenous variables, and often limited availability of high-quality labeled future data for supervision. The core objective is to learn a temporal extrapolation operator that maps a historical observation into a future prediction horizon.

Let $\boldsymbol{X} \in \mathbb{R}^{T \times V}$ denote the observed historical sequence, where $T \in \mathbb{N}^+$ is the input sequence length and $V \in \mathbb{N}^+$ is the number of variables. The goal is to learn a parametric mapping function:

$$\boldsymbol{F}_{\boldsymbol{\Theta_F}} : \mathbb{R}^{T \times V} \to \mathbb{R}^{T' \times V}$$

It is parameterized by $\boldsymbol{\Theta_F} \in \mathbb{R}^d$, and predicts the future trajectory $\hat{\boldsymbol{X}}_{[T:T+T')} \in \mathbb{R}^{T' \times V}$ over the next $T' \in \mathbb{N}^+$ time steps. Therefore, given a collection of $K$ distinct time series distributions $\{\mathcal{D}_1, \mathcal{D}_2, \ldots, \mathcal{D}_K\}$, and a differentiable loss function $\mathcal{L} : \mathbb{R}^{T' \times V} \times \mathbb{R}^{T' \times V} \to \mathbb{R}_{\geq 0}$, the optimal parameters $\boldsymbol{\Theta_F^*}$ are obtained by minimizing the expected risk over both task distributions. To evaluate the performance, several standard metrics are employed. Let $\boldsymbol{Y} = \boldsymbol{X}_{[T:T+T')} \in \mathbb{R}^{T' \times V}$ denote the ground truth future values, and $\hat{\boldsymbol{Y}} = \hat{\boldsymbol{X}}_{[T:T+T')} \in \mathbb{R}^{T' \times V}$ the corresponding predictions. Define the element-wise error matrix $\boldsymbol{E} = \hat{\boldsymbol{Y}} - \boldsymbol{Y} \in \mathbb{R}^{T' \times V}$.

## B.2  EVALUATION METRICS

**Mean Absolute Error (MAE)**:

$$\text{MAE} = \frac{1}{T'V} \sum_{t=1}^{T'} \sum_{v=1}^{V} \left| \hat{Y}_{t,v} - Y_{t,v} \right|$$

**Mean Squared Error (MSE)**:

$$\text{MSE} = \frac{1}{T'V} \sum_{t=1}^{T'} \sum_{v=1}^{V} \left( \hat{Y}_{t,v} - Y_{t,v} \right)^2$$

**Mean Absolute Percentage Error (MAPE)** (assuming $Y_{t,v} \neq 0$):

$$\text{MAPE} = \frac{100\%}{T'V} \sum_{t=1}^{T'} \sum_{v=1}^{V} \left| \frac{\hat{Y}_{t,v} - Y_{t,v}}{Y_{t,v}} \right|$$

While MAE and MSE emphasize absolute and squared deviations respectively, and MAPE captures relative errors across scales. In this paper, we focus on MAE and MSE due to their stability and robustness, particularly in scenarios where values may approach zero.

## B.3  DIVERSE FORECASTING SCENARIOS

The practical utility of a forecasting model is critically determined by its data efficiency — its ability to deliver accurate predictions under varying levels of access to target-domain supervision. We formalize three canonical regimes along this spectrum: *full-shot*, *few-shot*, and *zero-shot* learning.

**Full-shot learning**  In the full-shot regime, the model is trained and evaluated on abundant samples from the same distribution $\mathcal{D}_{\text{target}}$. Let $\mathcal{S}_{\text{train}} = \{(\boldsymbol{X}^{(i)}, \boldsymbol{Y}^{(i)})\}_{i=1}^{N}$ with $N \to \infty$ (or sufficiently large) denote the training set drawn from $\mathcal{D}_{\text{target}}$. The objective is to learn parameters $\boldsymbol{\Theta_F}$ that minimize the empirical risk:

$$\boldsymbol{\Theta_F^*} = \arg\min_{\boldsymbol{\Theta_F}} \frac{1}{N} \sum_{i=1}^{N} \mathcal{L}\left(\boldsymbol{F}_{\boldsymbol{\Theta_F}}(\boldsymbol{X}^{(i)}), \boldsymbol{Y}^{(i)}\right)$$

Full-shot learning is the classical supervised learning setup and serves as the performance upper bound for data-scarce regimes.

**Few-shot learning**  In few-shot forecasting, only a small support set $\mathcal{S}_{\text{support}} = \{(\boldsymbol{X}^{(i)}, \boldsymbol{Y}^{(i)})\}_{i=1}^{M}$ from $\mathcal{D}_{\text{target}}$ is available for adaptation.  The model, pre-trained on auxiliary distributions

$\{\mathcal{D}_1, \ldots, \mathcal{D}_K\}$, must rapidly adapt its parameters to minimize the target risk:

$$\mathcal{R}_{\text{target}} = \mathbb{E}_{(\boldsymbol{X}, \boldsymbol{Y}) \sim \mathcal{D}_{\text{target}}} \left[ \mathcal{L} \big( \boldsymbol{F}_{\boldsymbol{\Theta}_{\boldsymbol{F}}}(\boldsymbol{X}), \boldsymbol{Y} \big) \right]$$

**Zero-shot learning**  In the most challenging zero-shot regime, *no samples* from $\mathcal{D}_{\text{target}}$ are available during training or adaptation ($\mathcal{S}_{\text{support}} = \emptyset$). The model must generalize purely from inductive biases learned across $\{\mathcal{D}_1, \ldots, \mathcal{D}_K\}$. we require:

$$\boldsymbol{F}_{\boldsymbol{\Theta}_{\boldsymbol{F}}} \text{ trained on } \bigcup_{k=1}^{K} \mathcal{D}_k \Rightarrow \text{low } \mathcal{R}_{\text{target}} \text{ on unseen } \mathcal{D}_{\text{target}}$$

These three regimes form a hierarchy of practical difficulty: full-shot provides the ideal baseline; few-shot tests rapid adaptability; zero-shot evaluates true generalization. A TS forecasting system claiming robustness must demonstrate competence across all three — a criterion increasingly adopted in modern benchmarks (e.g., TIME-LLM, LTSF-LINEAR, CROSSFORMER).

# C  Time Series Modeling

## C.1  Time Series Embed

In TS forecasting, observations $\boldsymbol{X} \in \mathbb{R}^{T \times V}$ are rarely fed directly into architectures. Instead, they are first projected into a latent embedding space $\mathbb{R}^d$ via a learnable or fixed encoding function $\Phi_{\text{embed}}$, producing a tokenized representation $\boldsymbol{E} \in \mathbb{R}^{N \times d}$. Three principal embedding paradigms have emerged: **point-wise**, **patch-wise**, and **variable-wise embeddings**, each offering distinct inductive biases and trade-offs (Liu et al., 2024b; Nie et al., 2023; Kitaev et al., 2020).

**Point-wise Embedding.**  In this classical approach, each time step $t$ and each variable $v$ is embedded into a $d$-dimensional vector. For each $(t, v)$, a projection $\phi_{\text{point}} : \mathbb{R} \to \mathbb{R}^d$ is:

$$\boldsymbol{e}_{t,v} = \phi_{\text{point}}(X_{t,v}) = \boldsymbol{W}_{\text{emb}} \cdot X_{t,v} + \boldsymbol{b}_{\text{emb}}$$

While simple and expressive, point-wise embedding suffers from high token count ($N = T \cdot V$), leading to quadratic attention complexity and difficulty capturing temporal locality.

**Variable-wise Embedding.**  In multivariate settings, variable-wise embedding treats each channel as an independent "token sequence" across time. For variable $v$, the temporal slice $\boldsymbol{X}_{:,v} \in \mathbb{R}^T$ is embedded into a sequence of $T$ tokens via a variable-specific projection:

$$\boldsymbol{e}_{:,v} = \phi_{\text{var}}^{(v)}(\boldsymbol{X}_{:,v}) = \boldsymbol{W}_{\text{var}}^{(v)} \cdot \boldsymbol{X}_{:,v} + \boldsymbol{b}_{\text{var}}^{(v)},$$

This paradigm, used in models like iTransformer and Crossformer, enables modeling inter-variable dependencies while preserving temporal structure per channel.

**Patch-wise Embedding.**  To enhance local context modeling, patch-wise embedding groups consecutive time steps into "patches" before embedding. Let $P$ denote the patch size, and assume $T$ is divisible by $P$ for simplicity. The sequence is partitioned into $T/P$ patches, each of size $P \times V$, and projected via a learnable linear layer:

$$\boldsymbol{e}_i = \phi_{\text{patch}}\left(\boldsymbol{X}_{[(i-1)P:iP),:}\right) = \boldsymbol{W}_{\text{patch}} \cdot \text{vec}\left(\boldsymbol{X}_{[(i-1)P:iP),:}\right) + \boldsymbol{b}_{\text{patch}}$$

With patch-wise embedding currently dominating long-sequence forecasting due to its balance of efficiency and expressiveness, we adopt Patch-wise Embedding in the present work.

## C.2  Transformer Blocks

The Transformer architecture fundamentally relies on attention mechanisms to model dependencies within and across sequences. We formalize three canonical attention mechanisms that underpin modern Transformer designs.

**Vanilla Self-Attention**  Given an embedded sequence $\boldsymbol{E} \in \mathbb{R}^{N \times d}$, the scaled dot-product attention is defined as:

$$\text{Attention}(\boldsymbol{Q}, \boldsymbol{K}, \boldsymbol{V}) = \text{softmax}\left(\frac{\boldsymbol{Q}\boldsymbol{K}^\top}{\sqrt{d}}\right)\boldsymbol{V}$$

where $\boldsymbol{Q} = \boldsymbol{E}\boldsymbol{W}^Q$, $\boldsymbol{K} = \boldsymbol{E}\boldsymbol{W}^K$, $\boldsymbol{V} = \boldsymbol{E}\boldsymbol{W}^V$ are linear projections, and $\boldsymbol{W}^Q, \boldsymbol{W}^K, \boldsymbol{W}^V \in \mathbb{R}^{d \times d_k}$. It allows unrestricted information flow across all time steps, making it suitable for encoding historical context, but unsuitable for autoregressive generation, as it violates temporal causality.

**Cross-Attention**  In encoder-decoder architectures, cross-attention enables the decoder to attend to encoded representations of the input sequence. Let $\boldsymbol{E}_{\text{enc}} \in \mathbb{R}^{N \times d}$ denote encoder outputs, and $\boldsymbol{E}_{\text{dec}} \in \mathbb{R}^{M \times d}$ denote decoder inputs:

$$\text{CrossAttention}(\boldsymbol{E}_{\text{dec}}, \boldsymbol{E}_{\text{enc}}) = \text{softmax}\left(\frac{\boldsymbol{Q}_{\text{dec}}\boldsymbol{K}_{\text{enc}}^\top}{\sqrt{d}}\right)\boldsymbol{V}_{\text{enc}}$$

where $\boldsymbol{Q}_{\text{dec}} = \boldsymbol{E}_{\text{dec}}\boldsymbol{W}^Q$, $\boldsymbol{K}_{\text{enc}} = \boldsymbol{E}_{\text{enc}}\boldsymbol{W}^K$, $\boldsymbol{V}_{\text{enc}} = \boldsymbol{E}_{\text{enc}}\boldsymbol{W}^V$. Cross-Attention enables conditional generation and is widely used in early Transformer forecasting models, but introduces architectural complexity and often suffers from error propagation during auto-regressive decoding.

**Causal Self-Attention.** To enable autoregressive modeling — where prediction at time $t$ depends only on $\{1, \ldots, t-1\}$, a causal mask $M \in \mathbb{R}^{N \times N}$ is applied:

$$\mathrm{CausalAttention}(Q, K, V) = \mathrm{softmax}\left(\frac{QK^\top}{\sqrt{d}} + M\right)V$$

Causal attention is the cornerstone of *decoder-only* Transformers, which treat forecasting as a pure sequence completion task: the model is trained to predict future values conditioned on past context.

Recent advances in large-scale modeling — particularly in language and multimodal domains, have demonstrated the superiority of decoder-only architectures with causal attention. Following the trend in large-scale modeling, we adopt a decoder-only Transformer with causal self-attention for its simplicity, scalability, and strong empirical performance, enabling direct multi-step prediction without iterative error accumulation or encoder-decoder synchronization.

### C.3 PREDICTION HEAD

The hidden states from the final Transformer layer $H^{L-1} \in \mathbb{R}^{N \times d}$ are mapped to the target forecasting space through a linear Prediction Head. Let $W_{\mathrm{out}} \in \mathbb{R}^{d \times D}$ denote the learnable projection matrix, where $D = T' \cdot V$ is the total dimensionality of the prediction horizon, and $Y \in \mathbb{R}^{T' \times V}$ represents the final predictions:

$$Y = \mathrm{reshape}\left(\mathrm{LN}(H^{L-1})W_{\mathrm{out}}\right)$$

This direct multi-step mapping predicts all future time steps in a single forward pass, avoiding the error accumulation inherent in auto-regressive decoding. In our case, the use of a single-pass MLP further enhances efficiency by enabling lightweight computation and fast inference without sacrificing performance.

# D DATASETS

## D.1 DATASETS OF SINGLE-DATASET LEARNING

In the *single-dataset learning* setting, models are trained, validated, and tested on temporally contiguous splits drawn from a single TS distribution. It evaluates the model's **in-domain & full-shot learning capability**——learn and extrapolate temporal patterns when abundant target-domain data is available during training.

**ETT (Electricity Transformer Temperature)**: It includes four variants — ETTh1 and ETTh2 (hourly) and ETTm1 and ETTm2 (15-minutely). ETT are collected from two distinct regions over approximately two years. Each contains 7 variables, such as oil temperature, load, and equipment status and exhibit strong periodicity, trend drifts, and occasional abrupt shifts, making them standard benchmarks for long-term forecasting.

**Electricity**: Hourly electricity consumption records from 321 clients over four years. It features high volatility, heterogeneous load profiles, and complex inter-client correlations, challenging models to capture both global trends and individual behavioral dynamics.

**Exchange Rate**: Daily exchange rates of eight major currencies — Australia, UK, Canada, Switzerland, China, Japan, New Zealand, and Singapore, relative to the US dollar, spanning 16 years. It reflects slow macroeconomic trends, global financial coupling, and rare regime shifts, ideal for evaluating robustness under low-frequency non-stationarity.

**Solar**: 10-minute resolution solar power production from 137 stations in Alabama over one year. Dominated by strong diurnal cycles and weather-induced intermittency, it presents challenges in modeling fine-grained, spatially correlated renewable generation.

**Weather**: 21 meteorological variables, including air temperature, humidity, wind speed, and pressure, are recorded hourly at a German weather station over eight years. Exhibits rich multi-scale dynamics, suitable for evaluating hierarchical temporal modeling.

**Data Splitting Protocol.** To ensure consistency and avoid temporal leakage, all datasets are partitioned chronologically. For the four ETT datasets, we follow the established **8:4:4 month-wise split**: first 8 months for training, next 4 for validation, last 4 for testing. For all other datasets, we use a **7:1:2** ratio by sequence length: 70% training, 10% validation, 20% testing. All variables are standardized using training-set statistics only. No external covariates or calendar features are used, meaning models rely solely on historical observations.

## D.2 DATASETS OF CROSS-DATASET LEARNING & EVALUATION

In the *cross-dataset learning* setting, models are trained on the **union of training sets** from multiple distinct TS distributions, validated on the **union of validation sets**, and finally evaluated **separately on each dataset's test set**. This protocol evaluates a model's ability to learn *universal temporal representations* that generalize across domains. Unlike single-dataset learning (subsection D.1), this setup tests robustness to distribution shift, and parameter efficiency under heterogeneous data. We extend the dataset collection from subsection D.1 by including the Traffic dataset, resulting in a total of nine datasets for cross-dataset experiments.

**Traffic**: It contains hourly road occupancy rates, the percentage of time a road segment is occupied by vehicles, and are recorded across 862 sensors on San Francisco Bay Area freeways over two years. It exhibits complex spatio-temporal correlations, strong weekly seasonality, and sharp rush-hour peaks, making it a challenging benchmark for modeling large-scale, real-world urban dynamics.

During training, batches are sampled uniformly across datasets to ensure balanced exposure. At test time, metrics are computed *individually per dataset* to assess both in-domain and full-shot learning performance.

To further evaluate the **out-of-domain generalization** and **zero-shot learning performance** of models trained under the cross-dataset protocol, we introduce a held-out evaluation suite comprising eight additional TS datasets. These datasets are **never seen during training or validation** — their training and validation partitions are entirely withheld. Only their test sets are used for final evaluation,

providing a strict zero-shot benchmark that measures how well learned temporal representations transfer to completely unseen domains.

**NN5**: A collection of 111 daily TS representing sales records of non-food items from an anonymous retail chain in the UK, spanning 2 years. Characterized by intermittent demand, promotional spikes, and calendar-driven seasonality.

**PDB (Protein Data Bank)**: Weekly counts of new protein structure submissions to the PDB from 2000 to 2022. Exhibits slow, science-policy-driven growth trends with occasional abrupt shifts due to technological or institutional changes.

**Sceaux**: Hourly measurements of temperature, humidity, and pressure recorded at a weather station in Sceaux, France, over 5 years. Features fine-grained meteorological dynamics with strong diurnal and seasonal cycles.

**Smart**: Electricity consumption readings from 30 smart meters in Irish households over 18 months at 30-minute intervals. Captures heterogeneous household behavior, appliance-level patterns, and weather-sensitive usage.

**Spanish**: Daily electricity demand across Spain from 2014 to 2019. Reflects national-scale consumption modulated by economic activity, holidays, temperature, and renewable penetration.

**Sunspot-rain**: A bivariate dataset combining monthly international sunspot numbers and regional rainfall in Eastern Asia from 1850 to 2020. Offers an ultra-long-term, low-frequency view of potential solar-terrestrial coupling.

**USbirths**: Daily counts of births in the United States from 1969 to 1988. Displays strong weekly periodicity, holiday dips, and long-term demographic trends.

**Wind-power**: Hourly wind power generation from 20 turbines in a European wind farm over one year. Highly volatile and non-stationary, governed by weather dynamics and turbine-specific efficiency curves.

**Evaluation Protocol.** Critically, for all eight datasets above, **only the test set is used to test** and **no training or validation samples are exposed to the model at any stage**. It ensures a pure zero-shot evaluation: models must forecast these sequences based solely on representations learned from the original nine datasets.

The aforementioned 8+8 = 16 datasets will also be used for representation analysis, critical-layer localization, and pruning experiments on LLM4TS and TSFM families.

### D.3 DATASETS FOR BASELINE REPRESENTATION ANALYSIS AND PRUNING STUDIES

To evaluate the generality of our architectural findings, particularly in representation learning, critical layers localization and model compression, we extend our datasets with 4 additional traffic flow datasets from the **PEMS (Performance Measurement System) family**: **PEMS03**, **PEMS04**, **PEMS07**, and **PEMS08**. These datasets provide large-scale,w real-world transportation dynamics and serve as stress tests for structural robustness and efficiency under distribution shift.

**PEMS03**: Hourly traffic occupancy rates recorded by 358 sensors across the San Francisco Bay Area over 3 months in 2018. Exhibits strong spatial correlation and morning/evening rush-hour patterns.

**PEMS04**: Data from 307 sensors in the same region over 3 months in 2018, capturing similar dynamics to PEMS03 but with distinct sensor coverage and flow characteristics.

**PEMS07**: Records from 883 sensors in the Los Angeles metropolitan area over 6 months in 2017. Features higher spatial density and more complex congestion propagation patterns.

**PEMS08**: Measurements from 170 sensors in San Bernardino County over 6 months in 2016. Reflects suburban traffic dynamics with lower density but longer commute corridors.

**Data Splitting Protocol.** All four PEMS datasets follow a **7:1:2**, 70% for training, 10% for validation, and 20% for testing. Variables are standardized per dataset using training-set statistics only. No external features are included, ensuring models rely solely on learned temporal-spatial representations. PEMS are not used in cross-dataset learning or zero-shot evaluation.

# E    IMPLEMENTATION DETAILS

## E.1    MODEL FAMILY CONFIGURATIONS

Important architectural parameters are reported :Transformer layers, hidden channels, total learnable parameters, attention heads, positional encoding type, and embedding/prediction layers. All models use patch size 16 and stride 8 for time series embedding with linear input/output projections. We employ the commonly used setting of 336 input steps and 96 steps for future prediction.

| Models | | Transformer Blocks | | | | | | Time Seiries Embed | | | | Prediction Head | |
|---|---|---|---|---|---|---|---|---|---|---|---|---|---|
| Family | Scales | prior-para | layers | channels | learnable-para | heads | Positional Encoding | Input length | Patch Size | Stride | Input Layer | Output length | Output Layer |
| LLM4TS | Tiny | GPT-2 | 6 | 768 | 3.92M | 12 | learnable APE | 336 | 16 | 8 | Linear | 96 | Linear |
| | Small | GPT-2 | 12 | 768 | 3.93M | 12 | learnable APE | 336 | 16 | 8 | Linear | 96 | Linear |
| | Base | Qwen-3 | 28 | 1024 | 0.16B | 16 | RoPE | 336 | 16 | 8 | Linear | 96 | Linear |
| | Large | Qwen-3 | 28 | 2048 | 0.32B | 32 | RoPE | 336 | 16 | 8 | Linear | 96 | Linear |
| TSFMs | Tiny | RI | 6 | 768 | 85.02M | 12 | learnable APE | 336 | 16 | 8 | Linear | 96 | Linear |
| | Small | RI | 12 | 768 | 127.55M | 12 | learnable APE | 336 | 16 | 8 | Linear | 96 | Linear |
| | Base | RI | 28 | 1024 | 0.6B | 16 | RoPE | 336 | 16 | 8 | Linear | 96 | Linear |
| | Large | RI | 28 | 2048 | 1.73B | 32 | RoPE | 336 | 16 | 8 | Linear | 96 | Linear |

Table 10: Model Family Configurations. Model configurations across two families, LLM4TS and TSFMs, varying in scale from Tiny to Large.

We select two representative LLMs as backbones for the **LLM4TS** family: **GPT-2** (Radford et al., 2018) and **Qwen-3** (Yang et al., 2025). GPT-2 is a decoder-only Transformer pretrained on large-scale web text, employing learnable absolute positional embeddings and layer normalization before attention . Its simplicity, widespread adoption, and availability in multiple scales make it an ideal baseline for studying LLM adaptation to TS. Qwen-3 is a state-of-the-art Chinese-English bilingual LLM that adopts RoPE and RMSNorm, demonstrating superior long-context modeling and instruction-following capabilities. We include Qwen-3 to examine whether modern architectural advances, particularly relative position awareness and better scaling, and translate into improved TS forecasting performance. By repurposing LLMs without modifying their core architecture, we establish a strong, reproducible baseline for evaluating how well pretrained linguistic representations can be transferred to temporal forecasting.

## E.2    IMPLEMENTATION DETAILS FOR SUBSECTION 4.1 & 4.2

In subsection 4.1, all experiments are implemented using the `HuggingFace Transformers` library (v4.51.3) and trained with `DeepSpeed` (v0.14.0) and `Accelerate` (v0.28.0) for distributed training (Wolf et al., 2020). We use either $4\times$ NVIDIA A100 40GB GPUs or $4\times$ NVIDIA H100 80GB GPUs, depending on model scale. Training is conducted with a fixed random seed across all runs to ensure reproducibility. We employ the AdamW optimizer (Loshchilov & Hutter, 2019) with learning rate $1 \times 10^{-4}$, $\beta_1 = 0.9$, $\beta_2 = 0.95$, and $\epsilon = 1 \times 10^{-6}$. Weight decay is set to 0.01. The learning rate follows a cosine decay schedule without warmup. Training uses `bf16` mixed precision and DeepSpeed ZeRO Stage 2 for memory efficiency, with default gradient clipping. Gradient accumulation steps are set to 1, and micro batch size per GPU is fixed at 128. Early stopping is applied based on MSE of validation sets, with a patience of 3 epochs. All components are built on standard HuggingFace and PyTorch APIs.

In subsection 4.2, to investigate whether the performance advantages of advanced backbone architectures are contingent on abundant training data, we conduct a controlled data-scaling study. We vary the proportion of training data sampled from each dataset — from **20%** to **100%**, in increments of **20%**, while keeping all other experimental settings identical to subsection 4.1. Validation and test sets remain fixed and unchanged across runs to ensure consistent evaluation.

### E.3 Implementation Details for Subsection 4.3

In the cross-dataset learning setting, we combine the training partitions of all source datasets into a unified corpus. To avoid bias introduced by dataset ordering or temporal structure during training, we fully shuffle all samples globally before each epoch, ensuring that batches are statistically diverse and model updates are not dominated by any single domain. Due to the large volume of combined TS data and memory constraints, we preprocess and store all datasets in `Apache Arrow` format using the `datasets` library (v4.0.0), which enables efficient, zero-copy data loading and minimizes I/O bottlenecks.

All models are trained on $4\times$ NVIDIA H200 140GB GPUs, using the same codebase and hyper-parameters as in subsection 4.1, with the following specific adjustments: per-GPU batch size is set to 256, training is capped at a maximum of 3 epochs, and early stopping is triggered if no improvement in validation MSE is observed for 1 epoch. No curriculum learning, dataset weighting, or domain balancing strategy is applied. Models learn from uniformly sampled batches across all source domains, testing their raw capacity to acquire generalizable temporal representations under maximal data diversity.

### E.4 Implementation Details for Section 5.1

For four LLM4TS baselines in Section 5, we adopt **channel-independent strategy**, each variable is processed as an independent sequence. All reproduction and pruning experiments strictly adhere to same strategy and training hyperparameters reported in original papers, ensuring faithful re-implementation. Critically, the selection of trainable versus frozen modules is held identical across all models and experimental phases. For three TSFMs baselines, we freeze the entire Transformer backbone and update only the Time Series Embedding and Prediction Head modules during training. This isolates the contribution of architectural priors from optimization dynamics. To ensure a fair comparison between original models and their pruned counterparts, we maintain identical training strategies for both the originate models and the re-aligned pruned models . This eliminates confounding factors arising from training protocol differences and allows us to attribute performance changes solely to architectural modification.

# F  BASELINES

## F.1  BASELINES OF LLM4TS

FSCA (HU ET AL., 2025): Introducing Context-Alignment to comprehend TS with the same structural and logical awareness they apply to natural language. It achieves this through two complementary alignment mechanisms: (1) *Structural Alignment*, which employs dual-scale graph nodes to capture the hierarchical composition of TS, allowing LLMs to treat long sequences as coherent linguistic units while preserving fine-grained temporal features; and (2) *Logical Alignment*, which utilizes directed edges to model semantic dependencies across variables or time steps, ensuring contextual coherence and relational consistency.

CALF (LIU ET AL., 2025B): A dual-branch framework is employed, where aligned textual tokens and projected TS tokens are processed through the same frozen pretrained LLM layers to extract semantically aligned features. Cross-modal interaction is established via three alignment mechanisms: linear projection, cross-attention fusion, and contrastive tuning, enabling temporal forecasting grounded in linguistic priors without modifying the original LLM parameters.

TIME-LLM (JIN ET AL., 2023): The pretrained LLM is kept entirely frozen, and TS forecasting is enabled through two lightweight trainable modules, *Patch Reprogramming* for input adaptation and *Output Projection* for prediction mapping. It adopt a channel-independent strategy that decomposes multivariate forecasting into parallel univariate tasks.

OFA (ZHOU ET AL., 2023): A unified TS analysis framework is established by repurposing a frozen pre-trained language model without modifying its internal Transformer layers. By treating diverse TS tasks, including short- and long-term forecasting, classification, imputation, and anomaly detection, as sequence modeling problems within a common interface, the approach achieves highly competitive performance across all settings. Input TS are projected into the embedding space via lightweight trainable adapters, while outputs are mapped back through task-specific heads, preserving the LLM's original parameters throughout.

## F.2  BASELINES OF TSFMS

SUNDIAL (LIU ET AL., 2025C): A family of native TS foundation models is pre-trained on TimeBench, a corpus containing one trillion time points from real-world and synthetic datasets. Training employs a flow-matching-based TimeFlow Loss that enables direct optimization on continuous-valued sequences without discrete tokenization. By modeling the next-patch distribution in a non-parametric generative manner, the approach supports multi-modal probabilistic forecasting and effectively mitigates mode collapse. With only minimal adaptations to the Transformer architecture, the models accept arbitrary-length inputs, achieve strong scalability, and deliver state-of-the-art performance in both point and probabilistic forecasting, while enabling zero-shot inference in just a few milliseconds.

CHRONOS (ANSARI ET AL., 2024): It adapts TS to Transformer-based models through a two-step preprocessing pipeline of scaling and quantization. First, mean-scaling normalizes each value by the mean absolute magnitude of its historical context, ensuring comparability across series. Second, quantization discretizes scaled values into bins, yielding a sequence of discrete tokens. These tokens are modeled using a T5-based architecture trained with cross-entropy loss, enabling the acquisition of general-purpose TS representations while fully leveraging modeling paradigm.

TIMES-FM (DAS ET AL., 2024): As a patch-based forecasting framework, it supports variable context lengths and enables output patches longer than input patches. During training, model learns to predict extended horizons by conditioning on progressively longer prefixes. At inference time, long-term forecasting is performed in a semi-autoregressive manner: given a 256-step context, it predicts next 128 steps, then appends predictions to original context to forecast subsequent 128 steps. Internally, each decoding step processes current input patch through an input residual block, adds a positional encoding vector, feeds the result into a stacked Transformer with causal self-attention to ensure temporal consistency, and finally passes the representation through an output residual block to produce the output patch, which is compared against ground truth for loss computation.

# G    Full results

## G.1    Full results for Subsection 4.1

Table 11: Full results of performance of LLM4TS and TSFM families across different backbones.

| Models | | MAE | | | | | | | | | MSE | | | | | | | | |
| Family | Scales | ETTh1 | ETTh2 | ETTm1 | ETTm2 | Electricity | Exchange | Solar | Weather | avg | ETTh1 | ETTh2 | ETTm1 | ETTm2 | Electricity | Exchange | Solar | Weather | avg |
|---|---|---|---|---|---|---|---|---|---|---|---|---|---|---|---|---|---|---|---|
| LLM4TS | Tiny | 0.398 | 0.348 | 0.345 | 0.260 | 0.239 | 0.211 | 0.245 | 0.205 | **0.281** | 0.378 | 0.290 | 0.291 | 0.169 | 0.139 | 0.089 | 0.197 | 0.156 | **0.214** |
| | Small | 0.421 | 0.364 | 0.354 | 0.264 | 0.237 | 0.210 | 0.250 | 0.201 | **0.288** | 0.420 | 0.307 | 0.299 | 0.179 | 0.139 | 0.089 | 0.197 | 0.151 | **0.223** |
| | Base | 0.525 | 0.423 | 0.401 | 0.284 | 0.230 | 0.276 | 0.257 | 0.205 | **0.325** | 0.594 | 0.390 | 0.393 | 0.214 | 0.135 | 0.144 | 0.202 | 0.156 | **0.278** |
| | Large | 0.527 | 0.425 | 0.404 | 0.289 | 0.226 | 0.287 | 0.260 | 0.208 | **0.328** | 0.595 | 0.405 | 0.395 | 0.223 | 0.131 | 0.160 | 0.209 | 0.160 | **0.285** |
| TSFMs | Tiny | 0.430 | 0.360 | 0.355 | 0.264 | 0.227 | 0.223 | 0.235 | 0.206 | **0.288** | 0.420 | 0.310 | 0.304 | 0.177 | 0.133 | 0.096 | 0.188 | 0.153 | **0.223** |
| | Small | 0.450 | 0.364 | 0.358 | 0.265 | 0.228 | 0.230 | 0.235 | 0.207 | **0.292** | 0.464 | 0.307 | 0.310 | 0.184 | 0.133 | 0.094 | 0.183 | 0.154 | **0.229** |
| | Base | 0.469 | 0.384 | 0.372 | 0.268 | 0.228 | 0.238 | 0.252 | 0.211 | **0.303** | 0.482 | 0.316 | 0.334 | 0.185 | 0.135 | 0.133 | 0.239 | 0.162 | **0.248** |
| | Large | 0.478 | 0.388 | 0.375 | 0.271 | 0.234 | 0.253 | 0.252 | 0.212 | **0.308** | 0.506 | 0.325 | 0.333 | 0.183 | 0.140 | 0.141 | 0.242 | 0.164 | **0.254** |

Result shows enlarging the scale does not confer any performance advantage within each family, but results in performance degradation, shown in Table 11.

## G.2    Full results For Subsection 4.2

Table 12: Performance of LLM4TS families at Small and Base scales under varying training data ratios.

| Models | | MAE | | | | | | | | | MSE | | | | | | | | |
| Models | Ratio | ETTh1 | ETTh2 | ETTm1 | ETTm2 | Electricity | Exchange | Solar | Weather | avg | ETTh1 | ETTh2 | ETTm1 | ETTm2 | Electricity | Exchange | Solar | Weather | avg |
|---|---|---|---|---|---|---|---|---|---|---|---|---|---|---|---|---|---|---|---|
| LLM4TS-Small | 0.2 | 0.425 | 0.386 | 0.351 | 0.268 | 0.244 | 0.234 | 0.257 | 0.215 | **0.297** | 0.418 | 0.348 | 0.298 | 0.183 | 0.144 | 0.108 | 0.197 | 0.162 | **0.232** |
| | 0.4 | 0.439 | 0.365 | 0.356 | 0.267 | 0.234 | 0.217 | 0.247 | 0.218 | **0.293** | 0.445 | 0.320 | 0.305 | 0.179 | 0.139 | 0.092 | 0.195 | 0.168 | **0.230** |
| | 0.6 | 0.421 | 0.351 | 0.362 | 0.276 | 0.237 | 0.225 | 0.251 | 0.212 | **0.292** | 0.413 | 0.298 | 0.313 | 0.188 | 0.139 | 0.099 | 0.213 | 0.158 | **0.227** |
| | 0.8 | 0.419 | 0.360 | 0.361 | 0.261 | 0.237 | 0.221 | 0.253 | 0.204 | **0.290** | 0.411 | 0.307 | 0.308 | 0.174 | 0.139 | 0.095 | 0.194 | 0.153 | **0.222** |
| | 1.0 | 0.421 | 0.364 | 0.354 | 0.264 | 0.237 | 0.210 | 0.250 | 0.201 | **0.288** | 0.420 | 0.307 | 0.299 | 0.179 | 0.139 | 0.089 | 0.197 | 0.151 | **0.223** |
| LLM4TS-Base | 0.2 | 0.609 | 0.500 | 0.450 | 0.317 | 0.258 | 0.289 | 0.268 | 0.266 | **0.370** | 0.756 | 0.529 | 0.443 | 0.238 | 0.159 | 0.163 | 0.197 | 0.228 | **0.339** |
| | 0.4 | 0.595 | 0.459 | 0.447 | 0.334 | 0.241 | 0.269 | 0.282 | 0.266 | **0.362** | 0.742 | 0.450 | 0.444 | 0.254 | 0.144 | 0.140 | 0.199 | 0.225 | **0.325** |
| | 0.6 | 0.542 | 0.434 | 0.419 | 0.301 | 0.235 | 0.278 | 0.264 | 0.249 | **0.340** | 0.617 | 0.406 | 0.398 | 0.217 | 0.139 | 0.143 | 0.210 | 0.204 | **0.292** |
| | 0.8 | 0.533 | 0.433 | 0.420 | 0.304 | 0.232 | 0.272 | 0.270 | 0.227 | **0.336** | 0.600 | 0.415 | 0.413 | 0.215 | 0.138 | 0.138 | 0.207 | 0.179 | **0.288** |
| | 1.0 | 0.469 | 0.385 | 0.372 | 0.268 | 0.228 | 0.278 | 0.252 | 0.211 | **0.308** | 0.482 | 0.346 | 0.334 | 0.180 | 0.135 | 0.153 | 0.259 | 0.162 | **0.256** |

When the volume of training data increases, both MAE and MSE show a consistent downward trend, indicating that the models are able to learn more robust and generalizable representations. Interestingly, despite having a more advanced backbone, Larger-scale models do not always outperform their weaker counterparts under the same data regimes, shown in Table 12 & 13.

## G.3    Full results For Subsection 4.3

Full results of single-dataset learning are shown in Appendix G.1. Full results of in-domain performance in cross-dataset learning are shown in Table 14, and full results of out-of-domain performance in cross-dataset learning are shown in Table 15.

Table 13: Performance of TSFMs families at Tiny and Large scales under varying training data ratios.

| Models | Ratio | MAE | | | | | | | | | MSE | | | | | | | | |
|---|---|---|---|---|---|---|---|---|---|---|---|---|---|---|---|---|---|---|---|
| | | ETTh1 | ETTh2 | ETTm1 | ETTm2 | Electricity | Exechange | Solar | Weather | avg | ETTh1 | ETTh2 | ETTm1 | ETTm2 | Electricity | Exechange | Solar | Weather | avg |
| TSFMs-Tiny | 0.2 | 0.464 | 0.466 | 0.372 | 0.290 | 0.237 | 0.270 | 0.260 | 0.254 | **0.327** | 0.480 | 0.481 | 0.320 | 0.199 | 0.141 | 0.137 | 0.212 | 0.202 | **0.271** |
| | 0.4 | 0.446 | 0.388 | 0.360 | 0.284 | 0.232 | 0.236 | 0.255 | 0.221 | **0.302** | 0.453 | 0.337 | 0.308 | 0.193 | 0.137 | 0.113 | 0.207 | 0.171 | **0.240** |
| | 0.6 | 0.446 | 0.387 | 0.361 | 0.281 | 0.229 | 0.229 | 0.258 | 0.232 | **0.303** | 0.461 | 0.339 | 0.307 | 0.189 | 0.135 | 0.103 | 0.202 | 0.179 | **0.239** |
| | 0.8 | 0.490 | 0.390 | 0.357 | 0.286 | 0.228 | 0.238 | 0.247 | 0.218 | **0.307** | 0.521 | 0.345 | 0.305 | 0.192 | 0.134 | 0.108 | 0.204 | 0.167 | **0.247** |
| | 1.0 | 0.525 | 0.423 | 0.408 | 0.298 | 0.230 | 0.276 | 0.261 | 0.205 | **0.328** | 0.594 | 0.390 | 0.393 | 0.214 | 0.135 | 0.144 | 0.202 | 0.156 | **0.278** |
| TSFMs-Large | 0.2 | 0.499 | 0.417 | 0.409 | 0.290 | 0.244 | 0.311 | 0.273 | 0.238 | **0.335** | 0.524 | 0.385 | 0.380 | 0.208 | 0.149 | 0.188 | 0.202 | 0.184 | **0.277** |
| | 0.4 | 0.532 | 0.401 | 0.387 | 0.287 | 0.238 | 0.246 | 0.260 | 0.222 | **0.321** | 0.610 | 0.351 | 0.352 | 0.200 | 0.145 | 0.112 | 0.243 | 0.168 | **0.273** |
| | 0.6 | 0.478 | 0.373 | 0.364 | 0.273 | 0.233 | 0.242 | 0.246 | 0.230 | **0.305** | 0.499 | 0.326 | 0.317 | 0.186 | 0.139 | 0.113 | 0.209 | 0.184 | **0.247** |
| | 0.8 | 0.440 | 0.370 | 0.365 | 0.272 | 0.232 | 0.266 | 0.243 | 0.209 | **0.300** | 0.458 | 0.323 | 0.337 | 0.188 | 0.139 | 0.136 | 0.231 | 0.158 | **0.246** |
| | 1.0 | 0.478 | 0.373 | 0.361 | 0.271 | 0.234 | 0.243 | 0.252 | 0.209 | **0.303** | 0.506 | 0.325 | 0.333 | 0.183 | 0.140 | 0.111 | 0.240 | 0.164 | **0.250** |

Table 14: Full results of in-domain performance in cross-dataset learning.

| Models | | MAE | | | | | | | | | MSE | | | | | | | | |
|---|---|---|---|---|---|---|---|---|---|---|---|---|---|---|---|---|---|---|---|
| Family | Scales | ETTh1 | ETTh2 | ETTm1 | ETTm2 | Electricity | Exechange | Solar | Weather | avg | ETTh1 | ETTh2 | ETTm1 | ETTm2 | Electricity | Exechange | Solar | Weather | avg |
| LLM4TS | Small-C | 0.405 | 0.355 | 0.400 | 0.275 | 0.244 | 0.229 | 0.268 | 0.212 | **0.299** | 0.395 | 0.307 | 0.372 | 0.191 | 0.146 | 0.106 | 0.207 | 0.169 | **0.237** |
| | Base-C | 0.401 | 0.347 | 0.380 | 0.270 | 0.233 | 0.223 | 0.251 | 0.211 | **0.289** | 0.386 | 0.294 | 0.345 | 0.187 | 0.137 | 0.102 | 0.196 | 0.164 | **0.226** |
| | Large-C | 0.402 | 0.343 | 0.382 | 0.263 | 0.231 | 0.224 | 0.249 | 0.207 | **0.288** | 0.386 | 0.288 | 0.331 | 0.179 | 0.136 | 0.103 | 0.204 | 0.161 | **0.223** |
| TSFMs | Small-C | 0.401 | 0.350 | 0.365 | 0.265 | 0.228 | 0.225 | 0.256 | 0.209 | **0.287** | 0.389 | 0.305 | 0.320 | 0.182 | 0.134 | 0.101 | 0.214 | 0.161 | **0.226** |
| | Base-C | 0.435 | 0.358 | 0.377 | 0.275 | 0.232 | 0.239 | 0.256 | 0.218 | **0.299** | 0.447 | 0.313 | 0.352 | 0.195 | 0.140 | 0.110 | 0.265 | 0.175 | **0.250** |
| | Large-C | 0.421 | 0.370 | 0.379 | 0.278 | 0.234 | 0.240 | 0.261 | 0.221 | **0.301** | 0.424 | 0.338 | 0.355 | 0.201 | 0.142 | 0.114 | 0.284 | 0.177 | **0.254** |

Table 15: Full results of out-of-domain performance in cross-dataset learning.

| Models | | MAE | | | | | | | | | MSE | | | | | | | | |
|---|---|---|---|---|---|---|---|---|---|---|---|---|---|---|---|---|---|---|---|
| Family | Scales | NN5 | PDB | Sceaux | Smart | Spanish | Sunspot Rain | US Births | Wind Power | avg | NN5 | PDB | Sceaux | Smart | Spanish | Sunspot Rain | US Births | Wind Power | avg |
| LLM4TS | Small-C | 0.775 | 0.330 | 0.570 | 0.513 | 0.396 | 0.916 | 0.621 | 0.874 | **0.624** | 1.011 | 0.206 | 0.655 | 0.697 | 0.324 | 1.307 | 0.572 | 1.198 | **0.746** |
| | Base-C | 0.785 | 0.324 | 0.575 | 0.513 | 0.385 | 0.934 | 0.647 | 0.853 | **0.627** | 1.033 | 0.202 | 0.646 | 0.681 | 0.309 | 1.377 | 0.609 | 1.130 | **0.748** |
| | Large-C | 0.791 | 0.320 | 0.574 | 0.513 | 0.384 | 0.932 | 0.661 | 0.852 | **0.628** | 1.051 | 0.197 | 0.646 | 0.686 | 0.305 | 1.341 | 0.636 | 1.131 | **0.749** |
| TSFMs | Small-C | 0.786 | 0.322 | 0.587 | 0.546 | 0.382 | 1.018 | 0.630 | 0.879 | **0.644** | 1.017 | 0.199 | 0.653 | 0.714 | 0.300 | 1.580 | 0.604 | 1.154 | **0.777** |
| | Base-C | 0.804 | 0.339 | 0.623 | 0.569 | 0.413 | 0.976 | 0.653 | 0.918 | **0.662** | 1.100 | 0.218 | 0.725 | 0.790 | 0.347 | 1.401 | 0.630 | 1.307 | **0.815** |
| | Large-C | 0.827 | 0.350 | 0.628 | 0.575 | 0.412 | 1.037 | 0.683 | 0.942 | **0.682** | 1.156 | 0.236 | 0.741 | 0.821 | 0.345 | 1.607 | 0.671 | 1.402 | **0.872** |

While we might expect that exposure to a wider variety of TS patterns would enhance models' generalization ability, results suggest that both families are largely limited by other factors. Increasing the diversity of training data does not necessarily lead to corresponding performance improvements, neither on in-domain nor out-of-domain tasks, shown in Table 14 & 15. In some cases, even with more diverse data, models exhibit only marginal gains or plateau in performance, meaning simply expanding data diversity is insufficient to fully exploit the potential of backbones.

## G.4 FULL RESULTS FOR SUBSECTION 4.5

In Appendix G.1, we report in-domain performance of models under single-dataset learning. In Appendix G.3, we present in-domain and out-of-domain performance under cross-dataset learning. In this subsection, we report the full results obtained after extracting only a few Transformer layers and re-aligning them on the original training sets.

Table 16: Full results of performance of LLM4TS and TSFM families across different backbones(After Pruning).

| Models | | MAE | | | | | | | | | MSE | | | | | | | | |
| Family | Scales | ETTh1 | ETTh2 | ETTm1 | ETTm2 | Electricity | Exechange | Solar | Weather | avg | ETTh1 | ETTh2 | ETTm1 | ETTm2 | Electricity | Exechange | Solar | Weather | avg |
|---|---|---|---|---|---|---|---|---|---|---|---|---|---|---|---|---|---|---|---|
| LLM4TS | Small | 0.404 | 0.362 | 0.352 | 0.264 | 0.231 | 0.210 | 0.245 | 0.199 | **0.283** | 0.388 | 0.315 | 0.297 | 0.174 | 0.135 | 0.088 | 0.197 | 0.151 | **0.218** |
| | Base | 0.437 | 0.365 | 0.354 | 0.265 | 0.224 | 0.247 | 0.250 | 0.200 | **0.293** | 0.419 | 0.310 | 0.300 | 0.179 | 0.131 | 0.115 | 0.204 | 0.150 | **0.226** |
| | Large | 0.412 | 0.365 | 0.350 | 0.262 | 0.223 | 0.207 | 0.248 | 0.200 | **0.283** | 0.402 | 0.315 | 0.296 | 0.168 | 0.129 | 0.092 | 0.198 | 0.152 | **0.219** |
| TSFMs | Small | 0.420 | 0.362 | 0.360 | 0.262 | 0.226 | 0.224 | 0.235 | 0.203 | **0.287** | 0.417 | 0.309 | 0.298 | 0.170 | 0.136 | 0.098 | 0.184 | 0.154 | **0.221** |
| | Base | 0.406 | 0.358 | 0.364 | 0.266 | 0.227 | 0.221 | 0.229 | 0.205 | **0.284** | 0.410 | 0.303 | 0.301 | 0.177 | 0.132 | 0.101 | 0.180 | 0.158 | **0.220** |
| | Large | 0.436 | 0.362 | 0.359 | 0.274 | 0.217 | 0.212 | 0.237 | 0.221 | **0.290** | 0.427 | 0.316 | 0.292 | 0.182 | 0.130 | 0.133 | 0.202 | 0.166 | **0.231** |

Table 17: Full results of in-domain performance in cross-dataset learning(After Pruning).

| Models | | MAE | | | | | | | | | MSE | | | | | | | | |
| Family | Scales | ETTh1 | ETTh2 | ETTm1 | ETTm2 | Electricity | Exechange | Solar | Weather | avg | ETTh1 | ETTh2 | ETTm1 | ETTm2 | Electricity | Exechange | Solar | Weather | avg |
|---|---|---|---|---|---|---|---|---|---|---|---|---|---|---|---|---|---|---|---|
| LLM4TS | Small-C | 0.405 | 0.349 | 0.404 | 0.273 | 0.246 | 0.227 | 0.266 | 0.211 | **0.298** | 0.386 | 0.294 | 0.372 | 0.192 | 0.148 | 0.104 | 0.203 | 0.162 | **0.233** |
| | Base-C | 0.404 | 0.344 | 0.376 | 0.265 | 0.229 | 0.223 | 0.254 | 0.209 | **0.288** | 0.385 | 0.288 | 0.336 | 0.188 | 0.134 | 0.107 | 0.203 | 0.162 | **0.225** |
| | Large-C | 0.399 | 0.347 | 0.380 | 0.262 | 0.230 | 0.220 | 0.246 | 0.204 | **0.286** | 0.383 | 0.290 | 0.336 | 0.173 | 0.132 | 0.132 | 0.197 | 0.166 | **0.226** |
| TSFMs | Small-C | 0.409 | 0.349 | 0.366 | 0.263 | 0.228 | 0.230 | 0.242 | 0.205 | **0.286** | 0.386 | 0.306 | 0.321 | 0.185 | 0.137 | 0.104 | 0.214 | 0.161 | **0.227** |
| | Base-C | 0.415 | 0.354 | 0.371 | 0.274 | 0.234 | 0.225 | 0.254 | 0.217 | **0.293** | 0.426 | 0.308 | 0.354 | 0.190 | 0.141 | 0.109 | 0.273 | 0.175 | **0.247** |
| | Large-C | 0.423 | 0.361 | 0.271 | 0.269 | 0.224 | 0.236 | 0.249 | 0.224 | **0.282** | 0.431 | 0.314 | 0.194 | 0.183 | 0.137 | 0.124 | 0.231 | 0.177 | **0.224** |

Table 18: Full results of out-of-domain performance in cross-dataset learning(After Pruning).

| Models | | MAE | | | | | | | | | MSE | | | | | | | | |
| Family | Scales | NN5 | PDB | Sceaux | Smart | Spanish | Sunspot Rain | US Births | Wind Power | avg | NN5 | PDB | Sceaux | Smart | Spanish | Sunspot Rain | US Births | Wind Power | avg |
|---|---|---|---|---|---|---|---|---|---|---|---|---|---|---|---|---|---|---|---|
| LLM4TS | Small-C | 0.769 | 0.327 | 0.573 | 0.513 | 0.391 | 0.946 | 0.616 | 0.865 | **0.625** | 1.010 | 0.201 | 0.657 | 0.688 | 0.316 | 1.321 | 0.574 | 1.156 | **0.740** |
| | Base-C | 0.777 | 0.326 | 0.572 | 0.511 | 0.382 | 0.921 | 0.602 | 0.857 | **0.618** | 1.017 | 0.198 | 0.646 | 0.681 | 0.313 | 1.325 | 0.596 | 1.124 | **0.738** |
| | Large-C | 0.779 | 0.322 | 0.573 | 0.513 | 0.391 | 0.912 | 0.621 | 0.846 | **0.620** | 0.997 | 0.204 | 0.643 | 0.681 | 0.307 | 1.307 | 0.584 | 1.097 | **0.727** |
| TSFMs | Small-C | 0.790 | 0.315 | 0.594 | 0.531 | 0.374 | 0.975 | 0.645 | 0.856 | **0.635** | 1.106 | 0.196 | 0.679 | 0.700 | 0.287 | 1.511 | 0.612 | 1.163 | **0.782** |
| | Base-C | 0.814 | 0.345 | 0.615 | 0.536 | 0.425 | 0.971 | 0.642 | 0.885 | **0.654** | 0.987 | 0.249 | 0.666 | 0.702 | 0.346 | 1.412 | 0.615 | 1.206 | **0.773** |
| | Large-C | 0.830 | 0.337 | 0.622 | 0.541 | 0.385 | 0.984 | 0.629 | 0.946 | **0.659** | 1.143 | 0.222 | 0.722 | 0.724 | 0.298 | 1.456 | 0.605 | 1.396 | **0.821** |

Retaining only the critical Transformer layers, followed by fine-tuning on original training sets, not only preserves but in some cases even improves forecasting accuracy. Our approach effectively filters out less informative representations, allowing model to focus on the most salient temporal dependencies. In addition to maintaining predictive performance, it significantly reduces the total number of parameters, which in turn lowers memory footprint and accelerates inference.

### G.5 FULL RESULTS FOR SUBSECTION 5.1

**Baselines of LLM4TS**

We sequentially retain critical layers of four LLM4TS models—FSCA, CALF, TIME-LLM(G), and OFA, and perform re-alignment on the original training sets. In the following, we provide full results over horizons $\in \{96, 192, 336, 720\}$. We do not modify training strategies for the LLM4TS , retaining original experiment settings for either frozen or trainable parameters. For example, in the case of FSCA, original implementation fine-tuned the Layer-Norm and Word Positional Encoding, freezing MHSA and FFN to avoid catastrophic forgetting. We maintained this strategy for our experiments. Conversely, for TIME-LLM, the entire model was frozen during training, and we followed this same approach in our experiments.

Table 19: Full results of FSCA. Input length is 512, horizons $\in \{96, 192, 336, 720\}$

| Horizons | 96 | | | | 192 | | | | 336 | | | | 720 | | | | avg | | | |
|---|---|---|---|---|---|---|---|---|---|---|---|---|---|---|---|---|---|---|---|---|
| Metric | Pruned | | Original | | Pruned | | Original | | Pruned | | Original | | Pruned | | Original | | Pruned | | Original | |
| | MAE | MSE | MAE | MSE | MAE | MSE | MAE | MSE | MAE | MSE | MAE | MSE | MAE | MSE | MAE | MSE | MAE | MSE | MAE | MSE |
| **ETTh1** | 0.404 | 0.372 | 0.397 | 0.365 | 0.436 | 0.415 | 0.433 | 0.418 | 0.447 | 0.434 | 0.451 | 0.442 | 0.484 | 0.484 | 0.496 | 0.494 | 0.443 | 0.426 | 0.444 | 0.430 |
| **ETTh2** | 0.342 | 0.277 | 0.346 | 0.284 | 0.384 | 0.348 | 0.380 | 0.340 | 0.403 | 0.367 | 0.402 | 0.367 | 0.432 | 0.403 | 0.433 | 0.401 | 0.390 | 0.349 | 0.390 | 0.348 |
| **ETTm1** | 0.348 | 0.289 | 0.349 | 0.291 | 0.376 | 0.333 | 0.377 | 0.334 | 0.394 | 0.362 | 0.396 | 0.365 | 0.421 | 0.418 | 0.425 | 0.419 | 0.385 | 0.350 | 0.387 | 0.352 |
| **ETTm2** | 0.256 | 0.168 | 0.262 | 0.176 | 0.298 | 0.227 | 0.297 | 0.222 | 0.336 | 0.280 | 0.339 | 0.281 | 0.384 | 0.357 | 0.385 | 0.357 | 0.319 | 0.258 | 0.320 | 0.259 |
| **Electricity** | 0.240 | 0.138 | 0.237 | 0.134 | 0.251 | 0.151 | 0.252 | 0.152 | 0.268 | 0.164 | 0.271 | 0.169 | 0.306 | 0.208 | 0.306 | 0.209 | 0.266 | 0.165 | 0.267 | 0.166 |
| **Exchange** | 0.202 | 0.087 | 0.213 | 0.090 | 0.329 | 0.205 | 0.342 | 0.222 | 0.453 | 0.381 | 0.462 | 0.396 | 0.751 | 1.070 | 0.779 | 1.092 | 0.434 | 0.436 | 0.449 | 0.450 |
| **Soalr** | 0.255 | 0.192 | 0.260 | 0.195 | 0.254 | 0.190 | 0.266 | 0.204 | 0.270 | 0.201 | 0.284 | 0.222 | 0.277 | 0.210 | 0.286 | 0.220 | 0.264 | 0.198 | 0.274 | 0.210 |
| **Traffic** | 0.258 | 0.355 | 0.273 | 0.369 | 0.269 | 0.377 | 0.274 | 0.382 | 0.279 | 0.391 | 0.283 | 0.397 | 0.302 | 0.438 | 0.308 | 0.440 | 0.277 | 0.390 | 0.284 | 0.397 |
| **Weather** | 0.204 | 0.150 | 0.205 | 0.151 | 0.243 | 0.195 | 0.246 | 0.195 | 0.297 | 0.267 | 0.285 | 0.249 | 0.331 | 0.319 | 0.338 | 0.322 | 0.269 | 0.233 | 0.268 | 0.229 |
| **PEMS03** | 0.241 | 0.137 | 0.246 | 0.138 | 0.255 | 0.153 | 0.258 | 0.158 | 0.265 | 0.179 | 0.268 | 0.171 | 0.295 | 0.216 | 0.306 | 0.218 | 0.264 | 0.171 | 0.270 | 0.171 |
| **PEMS04** | 0.315 | 0.405 | 0.330 | 0.409 | 0.334 | 0.436 | 0.346 | 0.441 | 0.354 | 0.465 | 0.360 | 0.468 | 0.383 | 0.525 | 0.395 | 0.536 | 0.346 | 0.458 | 0.358 | 0.463 |
| **PEMS07** | 0.197 | 0.089 | 0.223 | 0.104 | 0.213 | 0.104 | 0.233 | 0.121 | 0.227 | 0.119 | 0.245 | 0.133 | 0.246 | 0.147 | 0.269 | 0.163 | 0.221 | 0.115 | 0.242 | 0.130 |
| **PEMS08** | 0.316 | 0.438 | 0.326 | 0.434 | 0.335 | 0.485 | 0.340 | 0.488 | 0.352 | 0.534 | 0.364 | 0.528 | 0.376 | 0.584 | 0.383 | 0.586 | 0.345 | 0.510 | 0.353 | 0.509 |

Table 20: Full results of CALF. Input length is 96, horizons $\in \{96, 192, 336, 720\}$

| Horizons | 96 | | | | 192 | | | | 336 | | | | 720 | | | | avg | | | |
|---|---|---|---|---|---|---|---|---|---|---|---|---|---|---|---|---|---|---|---|---|
| Metric | Pruned | | Original | | Pruned | | Original | | Pruned | | Original | | Pruned | | Original | | Pruned | | Original | |
| | MAE | MSE | MAE | MSE | MAE | MSE | MAE | MSE | MAE | MSE | MAE | MSE | MAE | MSE | MAE | MSE | MAE | MSE | MAE | MSE |
| **ETTh1** | 0.391 | 0.376 | 0.394 | 0.380 | 0.422 | 0.413 | 0.421 | 0.427 | 0.440 | 0.471 | 0.443 | 0.473 | 0.472 | 0.483 | 0.477 | 0.505 | 0.431 | 0.436 | 0.434 | 0.446 |
| **ETTh2** | 0.331 | 0.286 | 0.335 | 0.290 | 0.381 | 0.357 | 0.384 | 0.367 | 0.419 | 0.397 | 0.423 | 0.414 | 0.432 | 0.408 | 0.437 | 0.421 | 0.391 | 0.362 | 0.395 | 0.373 |
| **ETTm1** | 0.340 | 0.315 | 0.344 | 0.317 | 0.371 | 0.368 | 0.375 | 0.372 | 0.391 | 0.396 | 0.394 | 0.400 | 0.432 | 0.468 | 0.436 | 0.474 | 0.384 | 0.387 | 0.387 | 0.391 |
| **ETTm2** | 0.253 | 0.166 | 0.251 | 0.174 | 0.294 | 0.229 | 0.295 | 0.239 | 0.338 | 0.295 | 0.333 | 0.297 | 0.395 | 0.381 | 0.392 | 0.398 | 0.320 | 0.268 | 0.318 | 0.277 |
| **Electricity** | 0.245 | 0.165 | 0.248 | 0.165 | 0.258 | 0.173 | 0.258 | 0.174 | 0.270 | 0.186 | 0.274 | 0.190 | 0.303 | 0.212 | 0.307 | 0.228 | 0.269 | 0.184 | 0.272 | 0.189 |
| **Exchange** | 0.201 | 0.084 | 0.201 | 0.084 | 0.299 | 0.178 | 0.298 | 0.177 | 0.437 | 0.361 | 0.424 | 0.344 | 0.714 | 0.891 | 0.708 | 0.877 | 0.413 | 0.378 | 0.408 | 0.370 |
| **Soalr** | 0.245 | 0.205 | 0.248 | 0.211 | 0.263 | 0.244 | 0.267 | 0.267 | 0.259 | 0.233 | 0.280 | 0.262 | 0.273 | 0.226 | 0.279 | 0.262 | 0.260 | 0.227 | 0.268 | 0.251 |
| **Traffic** | 0.272 | 0.425 | 0.275 | 0.439 | 0.274 | 0.401 | 0.279 | 0.452 | 0.284 | 0.416 | 0.285 | 0.467 | 0.305 | 0.455 | 0.304 | 0.501 | 0.284 | 0.424 | 0.286 | 0.465 |
| **Weather** | 0.210 | 0.174 | 0.215 | 0.179 | 0.252 | 0.220 | 0.254 | 0.223 | 0.285 | 0.273 | 0.295 | 0.279 | 0.336 | 0.353 | 0.345 | 0.355 | 0.271 | 0.255 | 0.277 | 0.259 |
| **PEMS03** | 0.393 | 0.341 | 0.395 | 0.348 | 0.419 | 0.394 | 0.423 | 0.398 | 0.378 | 0.335 | 0.380 | 0.340 | 0.412 | 0.383 | 0.427 | 0.407 | 0.401 | 0.363 | 0.406 | 0.373 |
| **PEMS04** | 0.505 | 0.763 | 0.510 | 0.766 | 0.529 | 0.810 | 0.534 | 0.815 | 0.474 | 0.709 | 0.479 | 0.714 | 0.523 | 0.807 | 0.530 | 0.816 | 0.508 | 0.772 | 0.513 | 0.778 |
| **PEMS07** | 0.401 | 0.375 | 0.404 | 0.388 | 0.416 | 0.427 | 0.430 | 0.432 | 0.365 | 0.333 | 0.370 | 0.341 | 0.396 | 0.403 | 0.418 | 0.411 | 0.394 | 0.384 | 0.405 | 0.393 |
| **PEMS08** | 0.479 | 0.760 | 0.491 | 0.777 | 0.518 | 0.856 | 0.522 | 0.862 | 0.463 | 0.762 | 0.485 | 0.811 | 0.493 | 0.702 | 0.525 | 0.899 | 0.488 | 0.770 | 0.506 | 0.837 |

Table 21: Full results of TIME-LLM(G). Input length is 512, horizons ∈ {96, 192, 336, 720}

| Horizons | 96 | | | | 192 | | | | 336 | | | | 720 | | | | avg | | | |
|---|---|---|---|---|---|---|---|---|---|---|---|---|---|---|---|---|---|---|---|---|
| | Pruned | | Original | | Pruned | | Original | | Pruned | | Original | | Pruned | | Original | | Pruned | | Original | |
| Metric | MAE | MSE | MAE | MSE | MAE | MSE | MAE | MSE | MAE | MSE | MAE | MSE | MAE | MSE | MAE | MSE | MAE | MSE | MAE | MSE |
| ETTh1 | 0.411 | 0.385 | 0.409 | 0.381 | 0.430 | 0.413 | 0.442 | 0.426 | 0.467 | 0.465 | 0.471 | 0.468 | 0.477 | 0.463 | 0.516 | 0.519 | 0.446 | 0.431 | 0.459 | 0.448 |
| ETTh2 | 0.355 | 0.291 | 0.357 | 0.298 | 0.387 | 0.354 | 0.404 | 0.368 | 0.428 | 0.393 | 0.426 | 0.394 | 0.463 | 0.433 | 0.452 | 0.419 | 0.408 | 0.368 | 0.410 | 0.370 |
| ETTm1 | 0.350 | 0.292 | 0.354 | 0.295 | 0.387 | 0.354 | 0.385 | 0.352 | 0.393 | 0.363 | 0.393 | 0.369 | 0.430 | 0.421 | 0.424 | 0.422 | 0.390 | 0.358 | 0.389 | 0.359 |
| ETTm2 | 0.268 | 0.179 | 0.266 | 0.173 | 0.306 | 0.232 | 0.314 | 0.243 | 0.339 | 0.284 | 0.346 | 0.299 | 0.395 | 0.376 | 0.395 | 0.380 | 0.327 | 0.268 | 0.330 | 0.274 |
| Electricity | 0.236 | 0.134 | 0.253 | 0.143 | 0.247 | 0.149 | 0.263 | 0.159 | 0.274 | 0.168 | 0.271 | 0.170 | 0.296 | 0.202 | 0.296 | 0.202 | 0.263 | 0.163 | 0.271 | 0.169 |
| Exchange | 0.218 | 0.094 | 0.207 | 0.086 | 0.353 | 0.228 | 0.338 | 0.219 | 0.476 | 0.418 | 0.492 | 0.446 | 0.746 | 1.002 | 0.754 | 1.012 | 0.448 | 0.436 | 0.448 | 0.441 |
| Soalr | 0.254 | 0.179 | 0.256 | 0.179 | 0.248 | 0.183 | 0.247 | 0.183 | 0.256 | 0.193 | 0.255 | 0.189 | 0.268 | 0.206 | 0.270 | 0.205 | 0.257 | 0.190 | 0.257 | 0.189 |
| Traffic | 0.262 | 0.361 | 0.268 | 0.365 | 0.273 | 0.382 | 0.273 | 0.381 | 0.283 | 0.399 | 0.290 | 0.402 | 0.299 | 0.436 | 0.307 | 0.442 | 0.279 | 0.395 | 0.284 | 0.398 |
| Weather | 0.202 | 0.151 | 0.204 | 0.152 | 0.243 | 0.194 | 0.242 | 0.192 | 0.284 | 0.249 | 0.286 | 0.248 | 0.334 | 0.318 | 0.333 | 0.319 | 0.266 | 0.228 | 0.266 | 0.228 |
| PEMS03 | 0.248 | 0.143 | 0.248 | 0.143 | 0.269 | 0.169 | 0.268 | 0.171 | 0.288 | 0.185 | 0.291 | 0.184 | 0.324 | 0.237 | 0.329 | 0.242 | 0.282 | 0.183 | 0.284 | 0.185 |
| PEMS04 | 0.352 | 0.436 | 0.358 | 0.438 | 0.364 | 0.459 | 0.367 | 0.461 | 0.377 | 0.481 | 0.377 | 0.482 | 0.411 | 0.551 | 0.414 | 0.556 | 0.376 | 0.482 | 0.379 | 0.484 |
| PEMS07 | 0.231 | 0.105 | 0.231 | 0.110 | 0.249 | 0.135 | 0.252 | 0.140 | 0.244 | 0.131 | 0.247 | 0.134 | 0.277 | 0.175 | 0.284 | 0.186 | 0.250 | 0.137 | 0.253 | 0.142 |
| PEMS08 | 0.335 | 0.446 | 0.362 | 0.472 | 0.382 | 0.527 | 0.381 | 0.529 | 0.384 | 0.536 | 0.395 | 0.566 | 0.406 | 0.598 | 0.408 | 0.610 | 0.377 | 0.527 | 0.386 | 0.544 |

Table 22: Full results of OFA. Input length is 336, horizons ∈ {96, 192, 336, 720}

| Horizons | 96 | | | | 192 | | | | 336 | | | | 720 | | | | avg | | | |
|---|---|---|---|---|---|---|---|---|---|---|---|---|---|---|---|---|---|---|---|---|
| | Pruned | | Original | | Pruned | | Original | | Pruned | | Original | | Pruned | | Original | | Pruned | | Original | |
| Metric | MAE | MSE | MAE | MSE | MAE | MSE | MAE | MSE | MAE | MSE | MAE | MSE | MAE | MSE | MAE | MSE | MAE | MSE | MAE | MSE |
| ETTh1 | 0.401 | 0.382 | 0.398 | 0.378 | 0.424 | 0.422 | 0.420 | 0.417 | 0.448 | 0.458 | 0.436 | 0.444 | 0.465 | 0.455 | 0.481 | 0.482 | 0.435 | 0.429 | 0.434 | 0.430 |
| ETTh2 | 0.352 | 0.297 | 0.348 | 0.290 | 0.401 | 0.372 | 0.397 | 0.367 | 0.416 | 0.386 | 0.427 | 0.400 | 0.441 | 0.408 | 0.439 | 0.406 | 0.403 | 0.366 | 0.403 | 0.366 |
| ETTm1 | 0.350 | 0.294 | 0.348 | 0.293 | 0.380 | 0.341 | 0.370 | 0.331 | 0.396 | 0.369 | 0.396 | 0.370 | 0.427 | 0.424 | 0.426 | 0.428 | 0.388 | 0.357 | 0.385 | 0.355 |
| ETTm2 | 0.263 | 0.172 | 0.260 | 0.169 | 0.297 | 0.226 | 0.300 | 0.226 | 0.346 | 0.287 | 0.344 | 0.288 | 0.394 | 0.375 | 0.396 | 0.377 | 0.325 | 0.265 | 0.325 | 0.265 |
| Electricity | 0.231 | 0.135 | 0.239 | 0.139 | 0.247 | 0.153 | 0.254 | 0.157 | 0.264 | 0.169 | 0.270 | 0.172 | 0.296 | 0.207 | 0.299 | 0.208 | 0.259 | 0.166 | 0.265 | 0.169 |
| Exchange | 0.212 | 0.088 | 0.211 | 0.089 | 0.305 | 0.175 | 0.309 | 0.185 | 0.429 | 0.346 | 0.461 | 0.397 | 0.688 | 0.928 | 0.729 | 0.969 | 0.408 | 0.384 | 0.428 | 0.410 |
| Soalr | 0.245 | 0.197 | 0.249 | 0.203 | 0.258 | 0.212 | 0.273 | 0.215 | 0.274 | 0.216 | 0.285 | 0.219 | 0.280 | 0.221 | 0.298 | 0.227 | 0.264 | 0.211 | 0.276 | 0.216 |
| Traffic | 0.259 | 0.371 | 0.287 | 0.396 | 0.269 | 0.398 | 0.291 | 0.413 | 0.274 | 0.406 | 0.303 | 0.428 | 0.299 | 0.443 | 0.312 | 0.452 | 0.275 | 0.404 | 0.298 | 0.422 |
| Weather | 0.199 | 0.151 | 0.205 | 0.156 | 0.243 | 0.196 | 0.245 | 0.198 | 0.248 | 0.250 | 0.286 | 0.251 | 0.337 | 0.324 | 0.337 | 0.325 | 0.257 | 0.230 | 0.268 | 0.232 |
| PEMS03 | 0.236 | 0.135 | 0.247 | 0.140 | 0.257 | 0.167 | 0.274 | 0.175 | 0.268 | 0.180 | 0.280 | 0.186 | 0.312 | 0.229 | 0.321 | 0.233 | 0.268 | 0.178 | 0.281 | 0.183 |
| PEMS04 | 0.327 | 0.417 | 0.345 | 0.431 | 0.347 | 0.448 | 0.365 | 0.463 | 0.364 | 0.478 | 0.370 | 0.482 | 0.409 | 0.565 | 0.422 | 0.577 | 0.362 | 0.477 | 0.375 | 0.488 |
| PEMS07 | 0.216 | 0.104 | 0.227 | 0.113 | 0.232 | 0.127 | 0.246 | 0.135 | 0.237 | 0.137 | 0.248 | 0.143 | 0.258 | 0.177 | 0.289 | 0.190 | 0.235 | 0.136 | 0.252 | 0.145 |
| PEMS08 | 0.333 | 0.458 | 0.348 | 0.467 | 0.342 | 0.457 | 0.378 | 0.539 | 0.377 | 0.561 | 0.394 | 0.575 | 0.383 | 0.564 | 0.428 | 0.649 | 0.359 | 0.510 | 0.387 | 0.557 |

**Baselines of TSFMs**

During pruning, we freeze of all Transformer layers for the three TSFM models: SUNDIAL, CHRONOS, and TIMESFM. In the following, we provide full results over different horizons.

Table 23: Full results of SUNDIAL. Input length is 336, horizons $\in \{96, 192, 336, 720\}$

| Horizons | 96 | | | | 192 | | | | 336 | | | | 720 | | | | avg | | | |
|---|---|---|---|---|---|---|---|---|---|---|---|---|---|---|---|---|---|---|---|---|
| Metric | Pruned | | Original | | Pruned | | Original | | Pruned | | Original | | Pruned | | Original | | Pruned | | Original | |
| | MAE | MSE | MAE | MSE | MAE | MSE | MAE | MSE | MAE | MSE | MAE | MSE | MAE | MSE | MAE | MSE | MAE | MSE | MAE | MSE |
| ETTh1 | 0.438 | 0.434 | 0.458 | 0.453 | 0.490 | 0.509 | 0.493 | 0.514 | 0.523 | 0.571 | 0.543 | 0.602 | 0.564 | 0.621 | 0.575 | 0.654 | 0.504 | 0.534 | 0.517 | 0.556 |
| ETTh2 | 0.383 | 0.335 | 0.389 | 0.351 | 0.428 | 0.422 | 0.434 | 0.428 | 0.448 | 0.446 | 0.452 | 0.441 | 0.466 | 0.465 | 0.480 | 0.477 | 0.431 | 0.417 | 0.439 | 0.424 |
| ETTm1 | 0.369 | 0.318 | 0.375 | 0.330 | 0.388 | 0.353 | 0.414 | 0.394 | 0.410 | 0.390 | 0.434 | 0.431 | 0.455 | 0.464 | 0.476 | 0.506 | 0.405 | 0.381 | 0.425 | 0.415 |
| ETTm2 | 0.280 | 0.198 | 0.292 | 0.207 | 0.325 | 0.267 | 0.333 | 0.267 | 0.370 | 0.334 | 0.370 | 0.333 | 0.438 | 0.441 | 0.439 | 0.445 | 0.353 | 0.310 | 0.359 | 0.313 |
| Electricity | 0.229 | 0.131 | 0.233 | 0.131 | 0.246 | 0.149 | 0.250 | 0.149 | 0.258 | 0.157 | 0.268 | 0.167 | 0.309 | 0.212 | 0.301 | 0.203 | 0.261 | 0.162 | 0.263 | 0.162 |
| Exchange | 0.230 | 0.103 | 0.242 | 0.116 | 0.333 | 0.210 | 0.349 | 0.235 | 0.458 | 0.398 | 0.478 | 0.438 | 0.726 | 0.960 | 0.775 | 1.105 | 0.437 | 0.418 | 0.461 | 0.473 |
| Soalr | 0.235 | 0.181 | 0.247 | 0.183 | 0.247 | 0.191 | 0.261 | 0.194 | 0.252 | 0.196 | 0.270 | 0.204 | 0.286 | 0.214 | 0.271 | 0.210 | 0.255 | 0.195 | 0.262 | 0.198 |
| Traffic | 0.271 | 0.385 | 0.289 | 0.394 | 0.280 | 0.405 | 0.296 | 0.413 | 0.285 | 0.417 | 0.304 | 0.428 | 0.302 | 0.448 | 0.321 | 0.461 | 0.285 | 0.414 | 0.302 | 0.424 |
| Weather | 0.211 | 0.160 | 0.215 | 0.165 | 0.253 | 0.207 | 0.256 | 0.210 | 0.294 | 0.262 | 0.296 | 0.265 | 0.348 | 0.340 | 0.350 | 0.344 | 0.276 | 0.242 | 0.279 | 0.246 |
| PEMS03 | 0.230 | 0.126 | 0.235 | 0.135 | 0.254 | 0.154 | 0.256 | 0.161 | 0.261 | 0.162 | 0.267 | 0.164 | 0.286 | 0.195 | 0.298 | 0.210 | 0.258 | 0.159 | 0.264 | 0.168 |
| PEMS04 | 0.314 | 0.404 | 0.325 | 0.413 | 0.345 | 0.424 | 0.352 | 0.431 | 0.354 | 0.452 | 0.364 | 0.465 | 0.396 | 0.537 | 0.400 | 0.542 | 0.352 | 0.454 | 0.360 | 0.463 |
| PEMS07 | 0.207 | 0.104 | 0.191 | 0.092 | 0.225 | 0.124 | 0.208 | 0.111 | 0.235 | 0.136 | 0.220 | 0.123 | 0.266 | 0.174 | 0.253 | 0.161 | 0.233 | 0.134 | 0.218 | 0.122 |
| PEMS08 | 0.325 | 0.443 | 0.325 | 0.434 | 0.341 | 0.493 | 0.347 | 0.497 | 0.357 | 0.525 | 0.363 | 0.532 | 0.402 | 0.582 | 0.401 | 0.616 | 0.356 | 0.511 | 0.359 | 0.520 |

Table 24: Full results of CHRONOS. Input length is 96, horizons $\in \{24, 48, 96\}$

| Horizons | 24 | | | | 48 | | | | 96 | | | | avg | | | |
|---|---|---|---|---|---|---|---|---|---|---|---|---|---|---|---|---|
| Metric | Pruned | | Original | | Pruned | | Original | | Pruned | | Original | | Pruned | | Original | |
| | MAE | MSE | MAE | MSE | MAE | MSE | MAE | MSE | MAE | MSE | MAE | MSE | MAE | MSE | MAE | MSE |
| ETTh1 | 0.354 | 0.388 | 0.377 | 0.410 | 0.396 | 0.443 | 0.420 | 0.479 | 0.435 | 0.578 | 0.487 | 0.593 | 0.395 | 0.470 | 0.428 | 0.494 |
| ETTh2 | 0.258 | 0.206 | 0.269 | 0.200 | 0.305 | 0.250 | 0.314 | 0.268 | 0.354 | 0.328 | 0.376 | 0.342 | 0.306 | 0.261 | 0.320 | 0.270 |
| ETTm1 | 0.436 | 0.557 | 0.433 | 0.571 | 0.528 | 0.678 | 0.541 | 0.707 | 0.585 | 0.756 | 0.580 | 0.752 | 0.516 | 0.664 | 0.518 | 0.677 |
| ETTm2 | 0.236 | 0.144 | 0.221 | 0.136 | 0.255 | 0.181 | 0.271 | 0.189 | 0.339 | 0.281 | 0.365 | 0.294 | 0.277 | 0.202 | 0.286 | 0.207 |
| Electricity | 0.276 | 0.215 | 0.271 | 0.210 | 0.284 | 0.246 | 0.306 | 0.260 | 0.384 | 0.324 | 0.381 | 0.323 | 0.315 | 0.262 | 0.319 | 0.264 |
| Exchange | 0.112 | 0.031 | 0.108 | 0.026 | 0.152 | 0.044 | 0.150 | 0.048 | 0.209 | 0.115 | 0.224 | 0.110 | 0.158 | 0.063 | 0.161 | 0.061 |
| Soalr | 0.324 | 0.399 | 0.329 | 0.423 | 0.594 | 0.682 | 0.575 | 0.677 | 0.695 | 0.744 | 0.732 | 0.786 | 0.538 | 0.608 | 0.545 | 0.629 |
| Traffic | 0.366 | 0.519 | 0.373 | 0.585 | 0.435 | 0.674 | 0.434 | 0.664 | 0.456 | 0.729 | 0.475 | 0.763 | 0.419 | 0.641 | 0.428 | 0.671 |
| Weather | 0.137 | 0.125 | 0.130 | 0.123 | 0.194 | 0.186 | 0.192 | 0.178 | 0.237 | 0.210 | 0.250 | 0.215 | 0.189 | 0.174 | 0.191 | 0.172 |
| PEMS03 | 0.371 | 0.294 | 0.371 | 0.284 | 0.561 | 0.609 | 0.588 | 0.654 | 0.645 | 0.699 | 0.667 | 0.746 | 0.526 | 0.534 | 0.542 | 0.561 |
| PEMS04 | 0.380 | 0.407 | 0.395 | 0.416 | 0.566 | 0.654 | 0.565 | 0.656 | 0.632 | 0.706 | 0.695 | 0.735 | 0.526 | 0.589 | 0.552 | 0.602 |
| PEMS07 | 0.384 | 0.446 | 0.357 | 0.426 | 0.584 | 0.648 | 0.575 | 0.650 | 0.670 | 0.836 | 0.674 | 0.854 | 0.546 | 0.643 | 0.535 | 0.643 |
| PEMS08 | 0.419 | 0.648 | 0.415 | 0.655 | 0.594 | 0.799 | 0.596 | 0.808 | 0.726 | 0.875 | 0.746 | 0.883 | 0.579 | 0.774 | 0.586 | 0.782 |

# H ADDITIONAL VISUALIZATIONS & ANALYSES

## H.1 REPRESENTATIONS UNDER CROSS-DATASET LEARNING

Because of space constraints, only averaged results across distributions in single-dataset experiments are shown in the main text. Representation heatmaps regarding cross-dataset experiments are reported in this subsection.

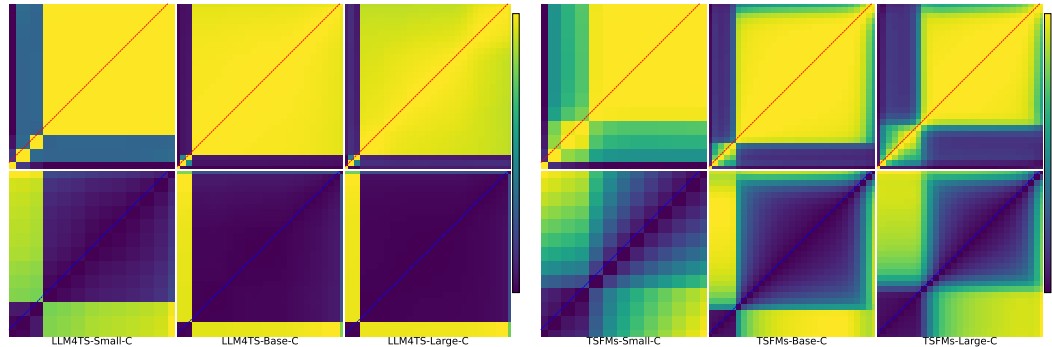

Figure 12: Inter-layer cosine similarity (*row 1*) and Euclidean distance (*row 2*). Brighter areas indicate higher values, darker areas lower values. While cross-dataset learning increases the diversity of data distributions, it remains insufficient for mitigating representation variations.

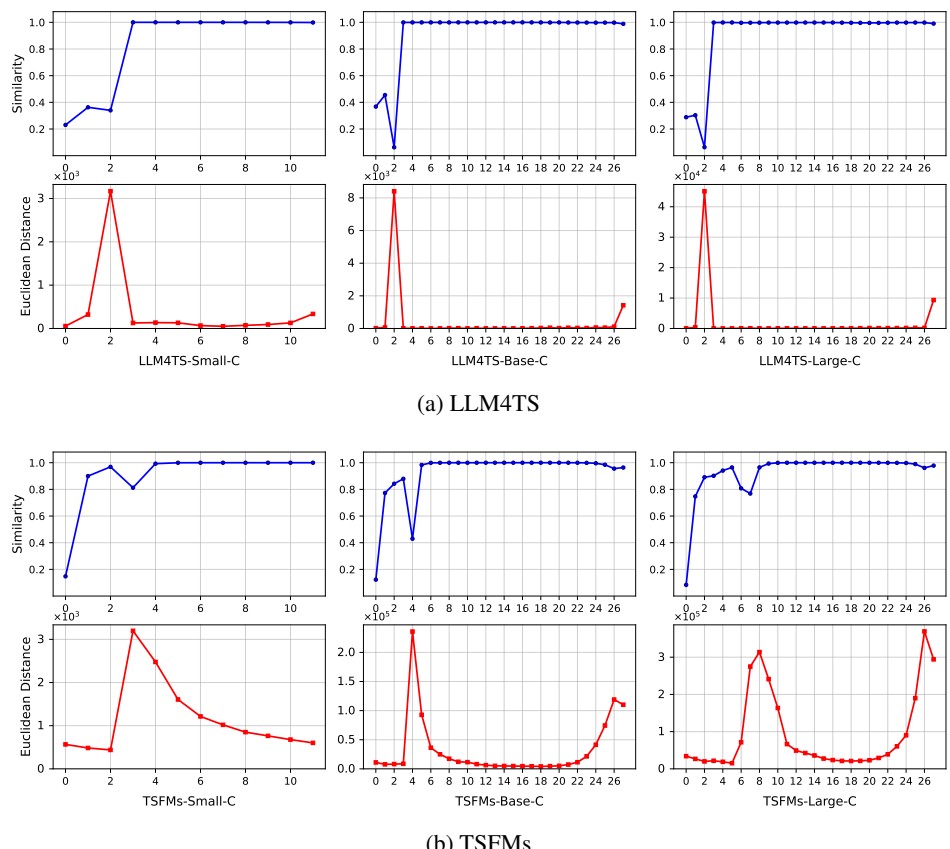

Figure 13: Per layer cosine similarity (*row 1*) and Per layer Euclidean distance (*row 2*).

Overall, the above results show cross-dataset learning brings little improvement in the activation of layers and heads, which helps explain why enriching data diversity does not enhance model vitality.

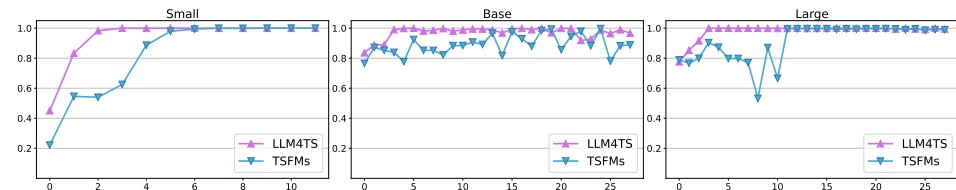

Figure 14: Average inter-layer pairwise similarity across all head attentions of cross-dataset learning.

The figure below provides supplementary results on representation from single-dataset learning, with a focus on capturing the relationship between representations and layers.

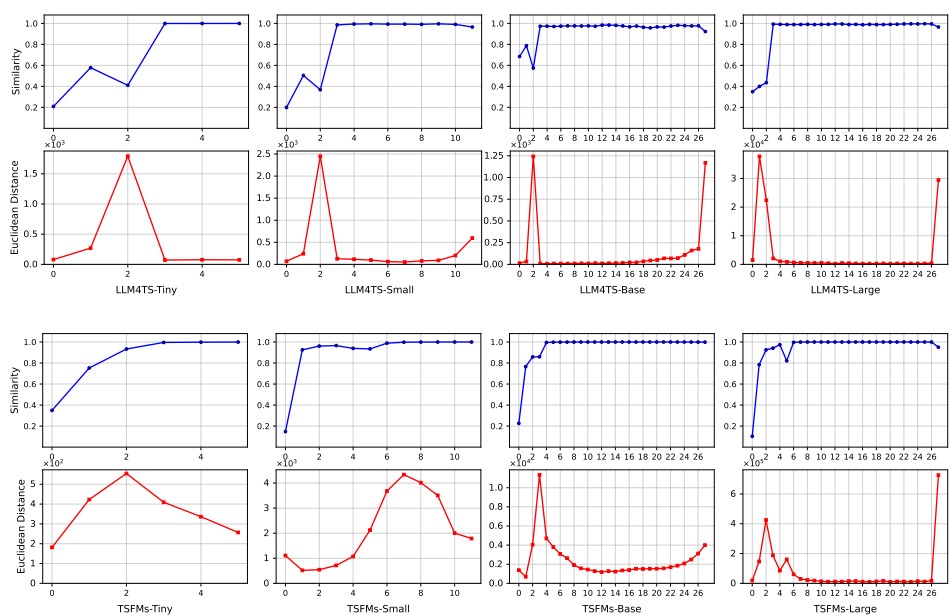

Figure 15: Inter-layer cosine similarity and Euclidean distance for LLM4TS and TSFMs.

## H.2 REPRESENTATION ANALYSIS OF BASELINES

Figure 17 shows the inter-layer representation similarity of Time-LLM (G) on Solar (forecasting 336 steps), Exchange (forecasting 336 steps), and ETTm2 with forecasting horizons of 96 and 720 steps. The effect of different prediction lengths and datasets on representation dynamics is minor.

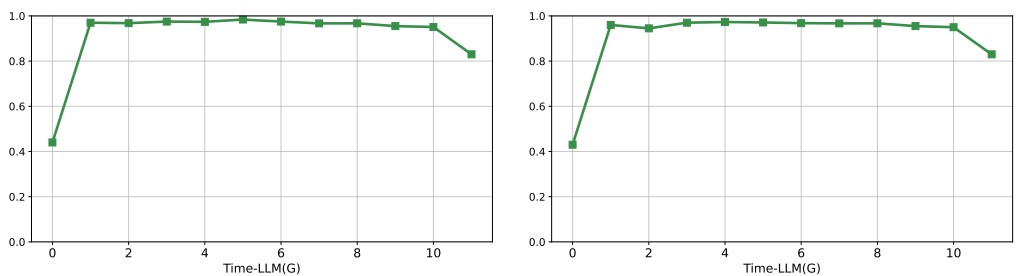

Figure 16: Representation patterns of TIME-LLM (G) on Solar and Exchange datasets.

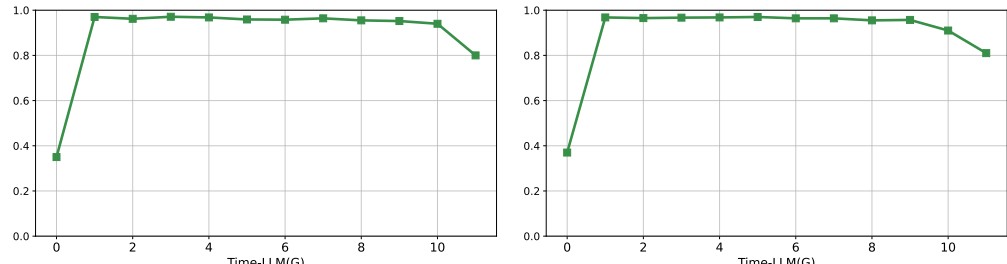

Figure 17: Representation patterns of TIME-LLM (G) on ETTm2 across different horizons.

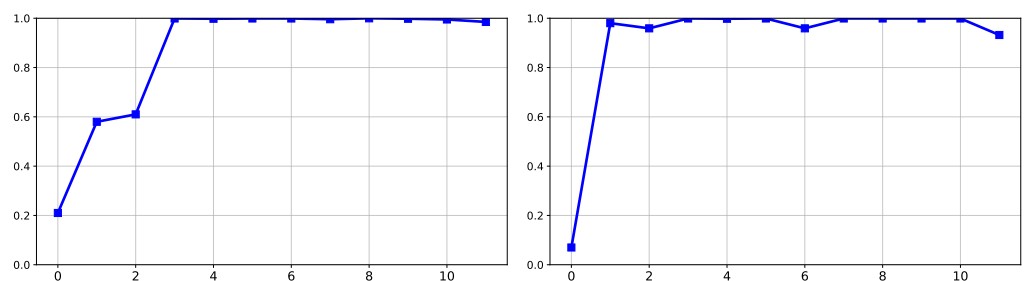

Figure 18: Representations patterns of FSCA on ETTh1 , left panel shows fully frozen text branch, and right panel depicts partially fine-tuned TS branch.

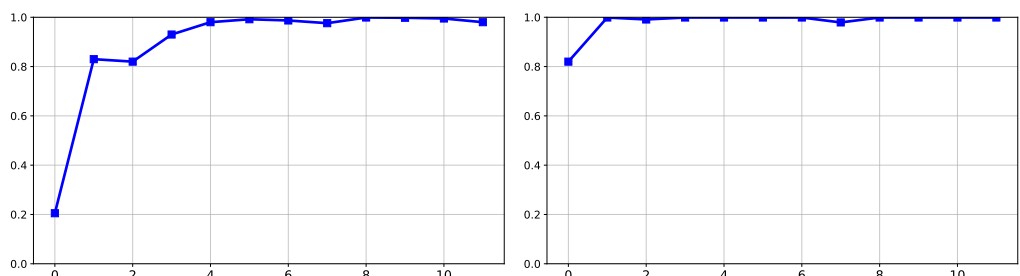

Figure 19: Representations patterns of FSCA on ETTm2 , left panel shows fully frozen text branch, and right panel depicts partially fine-tuned TS branch.

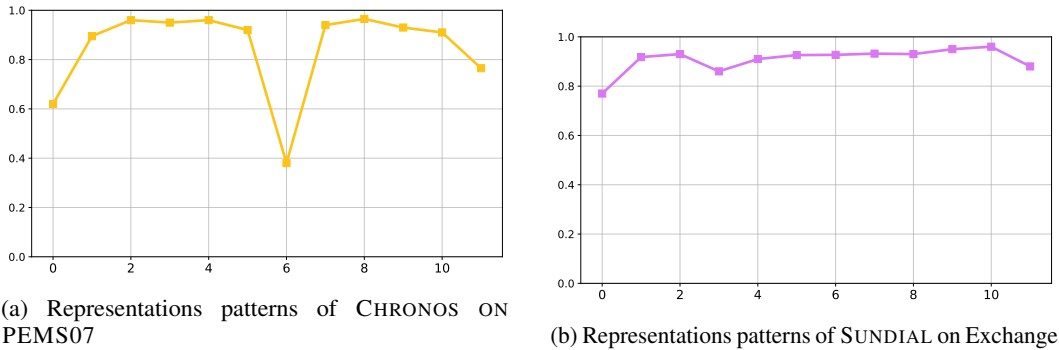

(a) Representations patterns of CHRONOS ON PEMS07

(b) Representations patterns of SUNDIAL on Exchange

Figure 20: Representation patterns of SUNDIAL and CHRONOS.

## H.3 FURTHER ANALYSIS

In subsection 7.1, taking the ETTh2 dataset as an example, the hidden states of each layer are sequentially passed through the Prediction Head to forecast a 96-step TS. Remarkably, when employing a train-free pruning strategy that conservatively retains only the first and last layers (column 3 & column 6), model still achieves comparatively strong inference performance. Furthermore, it is observed that directly projecting representations from intermediate layers back to TS leads to outputs that significantly deviate from true distribution (column 2 & column 5). (Column 1 & Column 4) correspond to the full model, where the outputs are obtained by feeding the representations from the initial state layer (Layer-init, representing the input state before any layer with positional encoding applied only) and the final layer into the Prediction Head. Further examples are shown below.

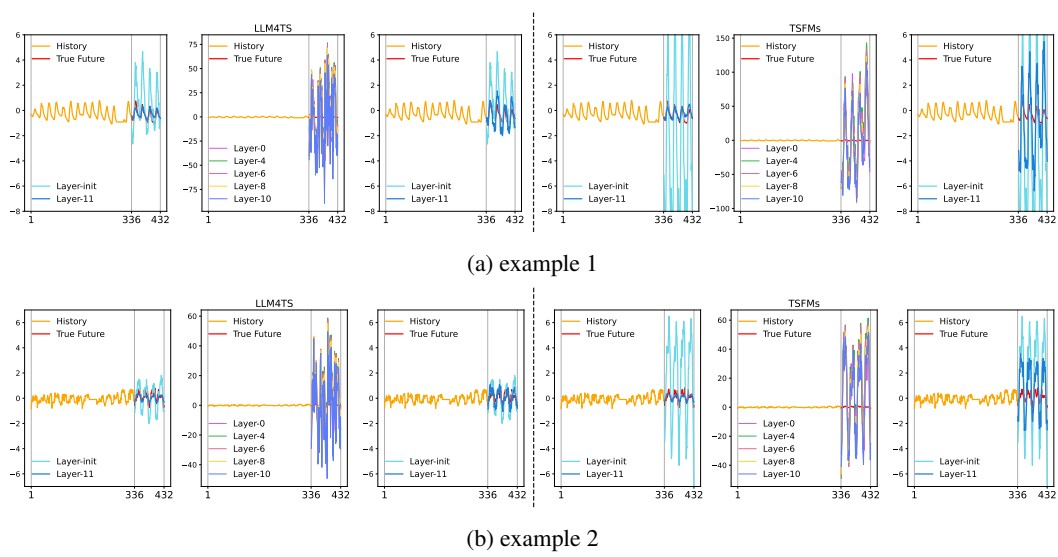

(a) example 1

(b) example 2

Figure 21: Keeping only first and last layers closely approaches the results of complete model.

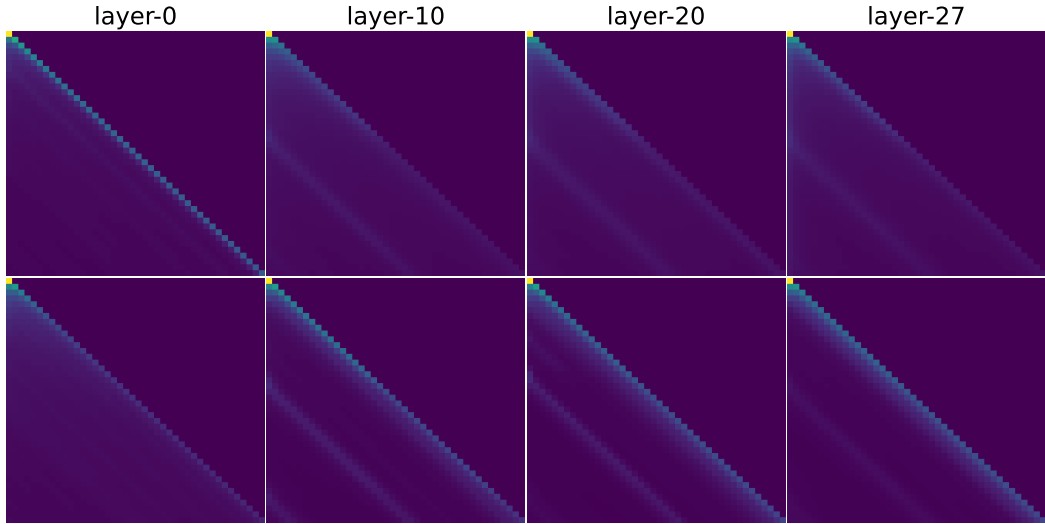

Figure 22: Similar to the observations in Subsection 7.2 for LLM4TS-Base and TSFMs-Base, the "Large" scale also exhibits a regular pattern of attention concentration, where tokens tend to attend primarily to themselves. Moreover, this effect is more pronounced in the TSFMs-Large (bottom) than in the LLM4TS-Large (top).

