# OpenReview forum: "The Few Govern the Many:Unveiling Few-Layer Dominance for Time Series Models"
_ICLR.cc/2026/Conference — Submitted to ICLR 2026_

### Official Review · Reviewer_E2is · 2025-10-31

**Soundness:** 2
**Presentation:** 1
**Contribution:** 3
**Rating:** 4
**Confidence:** 4

**Summary:**

This paper empirically investigates a counterintuitive “scaling paradox” in large-scale time series models and identifies a phenomenon termed “few-layer dominance,” in which only a small, critical subset of layers in a Transformer backbone actively contributes to learning temporal dynamics. Building on this finding, the authors propose a pruning method for large time series models that retains only the important layers, achieving performance comparable to the original models. The effectiveness of the proposed method is validated across seven large-scale time series models on multiple datasets. Moreover, the authors provide extensive ablation studies and additional experiments to further justify the soundness of the proposed pruning method.

**Strengths:**

1. The identification of “few-layer dominance” is a significant insight, and the proposed critical layer identification and pruning method also serves as an effective tool for model analysis and deployment.

2. The paper is well-motivated — it progresses logically from analyzing the scaling paradox to unveiling the few-layer dominance phenomenon and proposing a pruning approach. Each step is supported by experimental validation, making the overall work convincing.

3. In addition to providing sufficient experiments, the paper also offers clear takeaways that help readers understand the underlying mechanisms behind the observed phenomena.

**Weaknesses:**

1. The manuscript is not well formatted. Please revise the formatting and correct typo errors throughout the text.
- There are some issues with improper paragraph indentation, causing text overlap between lines.
- In Figure 1, the bottom row of the “Family” column in the right table should be “TSFMs.”
- Page 2, line 82: “stratch” should be corrected to “scratch.”
- Page 4, lines 183 and 205: “Appnedix” should be corrected to “Appendix.”
- Page 6, line 283: "XXX" is still a placeholder.
- Page 9, line 463: “thedegree” should be corrected to “the degree.

2. The proposed pruning method shows limited novelty, as the concept of pruning layers based on similarity-based importance metrics has been investigated in multiple prior studies.

[1] Pons I, Yamamoto B, Costa A H R, et al. Effective Layer Pruning Through Similarity Metric Perspective[C]//2nd Workshop on Advancing Neural Network Training: Computational Efficiency, Scalability, and Resource Optimization (WANT@ ICML 2024).

3. The experiments in Section 4 on large-scale time series (TS) models do not sufficiently justify the proposed takeaways and require more rigorous validation.
 -  Large TS models are typically TS foundation models (TSFMs) pretrained on massive datasets containing billions of observations. However, the datasets used in this study contain only about 18 million data points—far smaller than typical pretraining datasets. Therefore, the conclusions may be biased toward small-scale data regimes and cannot be generalized to large-scale settings.
- The effect of data diversity on both in-domain and out-of-domain performance is also insufficiently explored. The adopted datasets are primarily seasonal, meaning that the temporal pattern diversity does not significantly increase even when multiple datasets are used. Combined with the limited data volume, the models may lack strong out-of-domain generalization ability.
- Considering the limited training data, which may not sufficiently utilize the large model’s representation space, the takeaway of “few-layer dominance” might be overstated. The observed phenomenon could partly result from the underfitting of LLM4TS and TSFMs, rather than reflecting the behavior of real, pretrained-well TSFMs.

4. More comprehensive details of the experimental settings are needed to guarantee transparency and reproducibility. (Refer to the Questions section for a detailed list.)

**Questions:**

1. What are the fine-tuning settings when comparing the pruned models with the original models? Only under a fair and consistent fine-tuning strategy can the effectiveness of the pruning methods be properly evaluated.

2. The layer identification algorithm involves a hyperparameter, the decay factor $\alpha$. How is this hyperparameter set in the experiments, and how does it affect the effectiveness of pruning?

3. In Table 7, what is the purpose of comparing the pruned models with $\textbf{W}$(full LLM) ? It also appears that random pruning does not lead to a significant performance drop. Does this suggest that the model’s performance may be largely determined by the embedding layer, the output heads, and the fine-tuning process, rather than the pruned middle layers?

4. Is the observed “few-layer dominance” phenomenon common across other domains (e.g., CV and NLP), or does it reflect a limitation of the current Transformer architecture when applied to time series data?

---

### Official Review · Reviewer_MiPM · 2025-11-01

**Soundness:** 3
**Presentation:** 4
**Contribution:** 3
**Rating:** 4
**Confidence:** 4

**Summary:**

This paper presents the first systematic study of the scaling paradox in large Transformer-based time series models. Through extensive analysis, the authors reveal a few-layer dominance phenomenon -- only a handful of layers actively contribute to learning, while most remain redundant. By keeping only the most informative layers and fine-tuning them, pruned models maintain or even surpass the full models’ forecasting accuracy while achieving high efficiency.

**Strengths:**

S1 The paper introduces a concept -- few-layer dominance -- revealing that only a small subset of layers in large Transformer-based time series models actively contribute to learning. This reframes the scaling problem as layer laziness, providing an empirical view of why deeper architectures yield diminishing returns in time series modeling.

S2 The paper uses an analytical framework that jointly measures inter-layer representation shifts and intra-layer attention diversity to quantify each layer’s information. From this, a layer importance metric is used to assess how much each layer departs from lazy representation dynamics and contributes to meaningful feature transformation.

S3 Building on the analysis of layer laziness, the paper demonstrates that removing lazy layers and fine-tuning only the remaining active ones preserves and sometimes even improves forecasting accuracy.

S4: The research questions are clearly defined, and the conclusions (takeaway block) are clearly stated, supported by the corresponding experimental results.

**Weaknesses:**

W1: A large body of recent work has already investigated structural redundancy and layer importance in large Transformers. This paper primarily transfers those analytical methods to time-series models but does not provide domain-specific insights. Without a deeper connection to time-series dynamics (e.g., temporal correlation, seasonality, or frequency structure), the contribution is more like a direct application of existing LLM findings.

[1] Men, Xin, et al. "Shortgpt: Layers in large language models are more redundant than you expect." arXiv preprint arXiv:2403.03853 (2024).

[2] He, Shwai, et al. "What matters in transformers? not all attention is needed." arXiv preprint arXiv:2406.15786 (2024).


W2: Although the paper successfully identifies the phenomenon of layer laziness, it lacks a theoretical model that explains its underlying mechanism. The current analysis is largely empirical and descriptive, relying on measures of representation similarity and distance without establishing a causal explanation. In particular, the work does not clarify why time-series large Transformers are more susceptible to lazy representation dynamics than models in language or vision domains. An understanding that would be crucial for generalizing the findings beyond observation.


W3: The evaluation primarily compares three settings -- the full model, a randomly pruned version, and the proposed importance-based pruned model. While this validates that importance-guided pruning outperforms random selection, it does not fully isolate the role of layer laziness. Additional controls, such as training a model with the same reduced depth from scratch or fine-tuning only the identified critical layers while freezing others, would clarify whether the observed performance stems from genuine representational dynamics rather than from generic shallower optimization effects.

Suggestions: In Figure 2, different model scales (Tiny, Small, Base, Large) are distinguished only by line styles, which are difficult to differentiate visually. Using distinct bar patterns or textures would make the comparison across model sizes clearer.

**Questions:**

1. How would the conclusions change if the model were trained from scratch with the same reduced number of layers, instead of pruning a deeper network? If such shallow models perform similarly, would that imply that the large-scale parameterization is unnecessary for this kind of time series task? If not, could the authors provide possible explanations -- for instance, whether pretraining (especially for lazy layers) yields beneficial representation transfer?

2. Could the authors discuss whether layer laziness emerges from optimization dynamics or from task-level redundancy inherent in time series data?

---

### Official Review · Reviewer_HuEf · 2025-11-01

**Soundness:** 3
**Presentation:** 2
**Contribution:** 2
**Rating:** 2
**Confidence:** 4

**Summary:**

This paper investigates the “few-layer dominance” phenomenon in time-series forecasting models, arguing that only a small subset of layers substantially contributes to representation change and prediction quality. It proposes a heuristic layer-importance score to identify “key layers,” followed by a prune-and-realign (fine-tune) pipeline that aims to reduce inference cost while maintaining accuracy. Experiments across multiple TS architectures and scales support the empirical observation and show some efficiency gains.

**Strengths:**

- Clear empirical diagnosis of layer redundancy with simple, model-agnostic tooling (importance scoring + prune-and-realign).
- Broad benchmarking across architectures and sizes, showing consistent efficiency gains under certain settings.
- The proposed pipeline is practical to adopt and may serve as a useful diagnostic baseline for TS models.

**Weaknesses:**

- Numerous formatting issues (e.g., Line 221 overlapping lines; Line 463 “thedegree” missing a space); the paper needs thorough proofreading.
- The claim that “not all layers are equally important” has already been demonstrated in many tasks (e.g., [1][2]); moreover, time-series forecasting typically does not require large world-knowledge memory, so the contribution is limited in novelty.
- The technical contribution is relatively limited: the importance score is a heuristic composition with several hyperparameters (e.g., decay factor, K-nearest, top-τ filter, special-case exemptions), lacking comparisons against strong baselines (learnable layer gating, early-exit routing, distillation, etc.).
- Predictive performance cannot be consistently improved, and, in some cases (e.g., Table 23 on PEMS07), is clearly degraded.



[1] Lu, Yao, et al. "Reassessing layer pruning in llms: New insights and methods." arXiv preprint arXiv:2411.15558 (2024).

[2] Gromov, Andrey, et al. "The Unreasonable Ineffectiveness of the Deeper Layers." The Thirteenth International Conference on Learning Representations.

**Questions:**

- Do LLM architectures and their pre-trained knowledge genuinely offer greater potential for time-series forecasting, and if so, what are the concrete mechanisms (beyond capacity) that translate general linguistic/world knowledge into improved temporal representation and forecasting accuracy?
- Moreover, Table 5 reports LLM4TS-Small (GPT-2, 3.93M) outperforming LLM4TS-Large (Qwen3, 0.32B) in several settings; does this indicate that scaling LLM-style models for TS is largely meaningless?

---

### Official Review · Reviewer_rVu5 · 2025-11-02

**Soundness:** 3
**Presentation:** 2
**Contribution:** 3
**Rating:** 6
**Confidence:** 3

**Summary:**

This paper raises a new issue: in large models, bigger does not always mean better performance. The article also proves that only a few layers play a key role in the learning process. This is interest and meaningful work with reasonable experiments.

**Strengths:**

S1: This paper points out a problem in existing large-scale time series models: under both the TSFM and LLM4TS paradigms, a larger model does not mean better results.
S2: The paper explores multiple factors that influence model performance through three research questions: network scaling, data volume, and data homogeneity.
S3: The paper proposes a hypothesis that not all layers contribute significantly to the model's prediction performance; only a small number of layers dominate the final prediction. The article proves this hypothesis with extensive experiments。

**Weaknesses:**

W1: In research question 1 (Section 4.1), the article argues that scaling the backbone network does not improve performance. However, the author may have overlooked the issue of dataset stationarity. In Figure 2, the datasets where performance degrades as the model size increases (e.g., ETT, Exchange) are mostly those with poor stationarity or few data points. In this situation, increasing the network size may cause the model to overfit, reducing performance. In contrast, for datasets with high stationarity (Electricity, Weather), expanding the model size does not lead to a significant performance drop.
W2: Section 4.5 mentions pruning the model to keep only the most influential layers. The author should briefly describe the method. Furthermore, the necessity of pruning is also questionable. Based on the results in Figure 2, simply reducing the model size or adjusting it to a suitable size can achieve better performance.
W3: The representation of different model scales in Figure 2 is too vague and not intuitive.
W4: Most of the figures and tables in the article do not highlight the experiments that showed performance improvements. This makes it difficult to see at a glance which improvements were effective (for example, by marking models with improvements or the best performance in red).

**Questions:**

Q1: Can the author provide an analysis of dataset stationarity (considering W1)?
Q2: How is this pruning operation implemented (considering W2)?
Q3: Compared to adjusting the model size, what is the advantage of pruning (considering W2)?

---

### Meta-Review · Area_Chair_jYdH · 2026-01-05

**Summary:**

This paper investigates an interesting phenomenon in large-scale TS models, LLM4TS and TSFMs, that bigger is not always better, i.e., few-layer dominance.

All reviewers note that the core claim—only a subset of layers meaningfully contributes to model performance—has been extensively studied in prior work on Transformer redundancy and layer importance, especially in NLP/LLMs. The paper does not deliver too many fundamentally new methodological insights or techniques specific to the time-series domain.

The generality of the conclusions is still questionable. Reviewers raise concerns that the datasets used are relatively small compared to typical large-scale TS foundation models. Moreover, given that the paper title uses “Time Series Models,” the empirical analysis should not be only restricted to LLM4TS and TSFM. To support such a general claim, the phenomenon should also be examined on widely used classic deep time-series models (e.g., PatchTST, iTransformer).

While the paper is in an empirical style, reviewers consistently point out that the work lacks a convincing explanation of why this phenomenon happens in time-series models.

Multiple reviewers independently report substantial presentation problems, including formatting errors, unclear figures, and poorly described experimental details.

Last, the authors gave up the opportunity to do the rebuttal. Given the above, I recommend rejection.

**Reviewer Concerns:**

The authors did not do the rebuttal.

**Reviewer Scores:**

Reviewer rVu5 6->6, score will not change.

Reviewer HuEf 2->2, score will not change.

Reviewer MiPM 4->4, score will not change.

Reviewer E2is 4->4, score will not change.

---

### Decision · Program_Chairs · 2026-01-26

Reject